# The structure of a *Plasmodium vivax* Tryptophan Rich Antigen domain suggests a lipid binding function for a pan-*Plasmodium* multi-gene family

Prasun Kundu[1,6], Deboki Naskar[1,6], Shannon J. McKie[1], Sheena Dass[2], Usheer Kanjee[2], Viola Introini[1,3], Marcelo U. Ferreira[4,5], Pietro Cicuta [3], Manoj Duraisingh [2,7] ✉, Janet E. Deane [1,7] ✉ & Julian C. Rayner [1,7] ✉

Tryptophan Rich Antigens (TRAgs) are encoded by a multi-gene family found in all *Plasmodium* species, but are significantly expanded in *P. vivax* and closely related parasites. We show that multiple *P. vivax* TRAgs are expressed on the merozoite surface and that one, PVP01_0000100 binds red blood cells with a strong preference for reticulocytes. Using X-ray crystallography, we solved the structure of the PVP01_0000100 C-terminal tryptophan rich domain, which defines the TRAg family, revealing a three-helical bundle that is conserved across *Plasmodium* and has structural homology with lipid-binding BAR domains involved in membrane remodelling. Biochemical assays confirm that the PVP01_0000100 C-terminal domain has lipid binding activity with preference for sulfatide, a glycosphingolipid present in the outer leaflet of plasma membranes. Deletion of the putative orthologue in *P. knowlesi*, *PKNH_1300500*, impacts invasion in reticulocytes, suggesting a role during this essential process. Together, this work defines an emerging molecular function for the *Plasmodium* TRAg family.

*Plasmodium vivax* is the malaria parasite with the broadest geographic distribution, with nearly 3 billion people at risk of infection, primarily in Latin America and Asia[1]. It differs from *P. falciparum* in multiple aspects of its biology, most notably by the presence of the hypnozoite, a dormant liver stage that can lead to recurrent relapses even in the absence of transmission. In addition, the sexual stage gametocytes form much more rapidly, leading to more frequent transmission that is harder to control. As a result, *P. vivax*-specific intervention strategies are a defined goal for the WHO Malaria Control Plan[2]. However,

progress in understanding *P. vivax* biology has been slowed in part by its absolute requirement for growth in immature red blood cells called reticulocytes, which has hampered development of in vitro culture systems.

The process by which *Plasmodium* parasites identify and invade human red blood cells has long been of interest for vaccine development, as it is essential for parasite survival and therefore malaria pathology. In *P. falciparum*, the essential invasion protein PfRh5, which binds the erythrocyte receptor Basigin, is a high priority vaccine

[1]Cambridge Institute for Medical Research, University of Cambridge, Cambridge, UK. [2]Department of Immunology and Infectious Diseases, Harvard T.H. Chan School of Public Health, Boston, MA, USA. [3]Cavendish Laboratory, Department of Physics, University of Cambridge, Cambridge, UK. [4]Department of Parasitology, Institute of Biomedical Sciences, University of São Paulo, São Paulo, Brazil. [5]Global Health and Tropical Medicine, Associate Laboratory in Translation and Innovation Towards Global Health, LA-REAL, Institute of Hygiene and Tropical Medicine, NOVA University of Lisbon, Lisbon, Portugal. [6]These authors contributed equally: Prasun Kundu, Deboki Naskar. [9]These authors jointly supervised this work: Julian C. Rayner, Janet E. Deane, Manoj Duraisingh. ✉e-mail: mduraisi@hsph.harvard.edu; jed55@cam.ac.uk; jcr1003@cam.ac.uk

target[3,4] and Phase IIa trials show that even early-stage PfRh5 vaccines can reduce parasite growth rates in vivo[5]. The phylogenetic distance between *P. falciparum* and *P. vivax* means that there is no clear identifiable orthologue of PfRh5 in *P. vivax*, and blood-stage vaccine attention has focussed primarily on Duffy Binding Protein, PvDBP, with early stage vaccine trials under way[6]. However, sequence and copy number variation in PvDBP between *P. vivax* isolates is well-known[7] raising the potential for vaccine escape and emphasising that other blood-stage targets require investigation. Of particular interest are *P. vivax* proteins that specify reticulocytes for invasion. The recognition of reticulocytes depends, at least in part, on the interaction between *P. vivax* Reticulocyte Binding Protein 2b (PvRBP2b) and CD71/Transferrin Receptor[8], but the process of red blood cell invasion is a highly complex, multi-stage process[9], which likely involves additional proteins.

Potential candidate proteins include the Tryptophan rich antigen multi-gene family (TRAgs), which all share a Tryptophan-Threonine-rich domain of unknown function (PFAM ID PF12319, OrthoMCL ID OG6_145873), usually at their C-terminus. First identified in the *Plasmodium yoelii* (a species of parasite that causes malaria in rodents), TRAgs have been shown to generate highly protective antibodies in mice upon immunisation[10,11]. All *Plasmodium* species possess some TRAgs, but they are particularly numerous in *P. vivax* and closely related species, where they are frequently found in clusters along chromosomes, indicative of expansion through gene duplication and diversification. The precise function of TRAgs in not known in any *Plasmodium* species, but several members of the *P. vivax* TRAg family have been reported to have red blood cell binding properties[12] and in some cases interaction partners have been reported[13,14]. TRAgs also differ in expression between *P. vivax* infections in *Samiri* and *Aotus* monkeys, suggestive of a variable role, potentially as alternative invasion ligands to PvDBP[15]. Members of the *P. vivax* TRAg family are highly immunoreactive, with a mean seropositivity rate of around 60%, even in samples from low endemic regions[16].

In this work we explored the function of *P. vivax* TRAgs in a systematic manner using a eukaryotic recombinant protein expression system that we have previously employed to identify the interaction between PfRh5 and Basigin[3] and to screen for *P. vivax* vaccine candidates[17,18]. Antibodies raised against multiple *P. vivax* TRAgs suggested a merozoite surface localisation and revealed some limitations in current genome annotation. Focusing on one family member, PVP01_0000100, we establish that it binds to both erythrocytes and reticulocytes, with a preference for the latter, mediated via the C-terminal domain. Solving the crystal structure of this domain reveals an extended and curved three-helical bundle, with structural homology to lipid binding BAR domains. AlphaFold2 predictions suggest this core fold is widely conserved across *P. vivax* TRAgs and also TRAgs in other *Plasmodium* species. Lipid and liposome binding assays confirm that the PVP01_0000100 TRAg domain binds specifically to sulfatide and that other PvTRAgs also have lipid binding activity. Deletion of the

homologous gene in *P. knowlesi* results in a decreased preference for reticulocytes in invasion assays. This combination of biochemistry, structural biology and experimental genetics reveals a compelling molecular function for PVP01_0000100 and raises interesting hypotheses for the broader *Plasmodium* TRAg gene family.

## Results

### Tryptophan Rich Antigens (TRAgs) are expanded in some *Plasmodium* species

To facilitate a systematic exploration of the TRAg family we used three different methods to identify TRAgs: searching PFAM and OrthoMCL databases with the relevant domain ID for their conserved Tryptophan/Threonine-rich domain (PF12319 and OG6_145873 respectively) and BLAST searching PlasmoDB using the C-terminal domain from PVP01_0000100, eliminating low confidence hits. The numbers of TRAgs identified differed depending on the method used (Table 1), as the databases all use different reference genomes and different domain definitions. However, it is clear that the TRAg family, while present in all *Plasmodium* species, has expanded significantly in the primate *Plasmodium* clade that includes four of the five major human *Plasmodium* species−*P. vivax, P. knowlesi, P. ovale* and *P. malariae*. Multiple different nomenclatures have been used for the TRAg family, which complicates analysis and comparison[16,19]. To facilitate understanding we use the current *P. vivax* reference genome gene IDs throughout this work and provide a systematic summary of all previous nomenclature systems in Supplementary Table 1. A neighbour joining tree based on full-length TRAg sequences from *P. vivax, P. knowlesi* and *P. cynomolgi* (Supplementary Fig. 1) reveals that many genes have 1:1:1 orthologues in all three species, suggesting these TRAgs originated in a common ancestor before these species diverged.

There were however, examples of closely related genes in *P. vivax* and *P. cynomolgi* with no homologue present in *P. knowlesi*, suggesting they originated after the divergence of a common ancestor of *P. vivax* and *P. cynomolgi* from *P. knowlesi*. There are also *P. vivax* TRAgs with no clear 1:1 orthologue in either species, suggesting an origin after the split of *P. vivax* and *P. cynomologi*. This pattern suggests the process that led to the expansion of the TRAg family in the *Plasmodium* sub-genus is still occurring, with new genes evolving as *Plasmodium* species diverge.

### Certain *P. vivax* TRAgs are localised on the merozoite surface irrespective of signal peptide prediction

In keeping with previous analyses[16] some of the *P. vivax* TRAgs we identified are predicted to have Signal Peptides (SPs) in PlasmoDB, while some are not. However, all predictions in PlasmoDB are made using one SP prediction server, SignalP[20]. Analysis of *P. vivax* TRAg sequences using alternative servers, Phobius[21] and PredSci[22], identified potential SPs in several TRAgs that are not identified by SignalP (Supplementary Table 2). Notably, all *P. vivax* TRAgs lacking a predicted SP instead contained a domain annotated as 'N-terminal transmembrane', which we hypothesised may function as an SP. To test this, we generated polyclonal antibodies in rabbits against four PvTRAgs, both with and without predicted SPs. Antibodies were raised against recombinant proteins spanning the predicted ectodomain (ECD) (ie. from the end of the N-terminal hydrophobic region to the C-terminus of the protein, predicted using servers TMHMM 2.0 and Phobius; amino acid boundaries and sequences of all constructs are included in Supplementary Table 1 and Supplementary Table 3). These recombinant proteins were expressed in the HEK293E cell system, which has been used extensively for *Plasmodium* protein expression[23], and purified using affinity chromatography. To test for cross-reactivity, we used the antisera to probe Western blots containing a panel of 12 purified recombinant *P. vivax* TRAgs. Since all the constructs have a Biolinker peptide and hexa-His tag in common, some level of

**Table 1 | Expansion of the TRAgs in specific *Plasmodium* species**

| Organism | PFAM PF121319 | OrthoMCL OG6_145873 | PlasmoDB BLAST |
|---|---|---|---|
| *P. vivax* | 36 | 35 | 38 |
| *P. knowlesi* | 29 | 24 | 29 |
| *P. cynomolgi* | 31 | 33 | 35 |
| *P. ovale curtesi* | 43 | 32 | 40 |
| *P. malariae* | 27 | 25 | 35 |
| *P. falciparum* | 3 | 3 | 3 |
| *P. berghei* | 7 | 5 | 5 |
| *P. yoelii* | 5 | 5 | 5 |

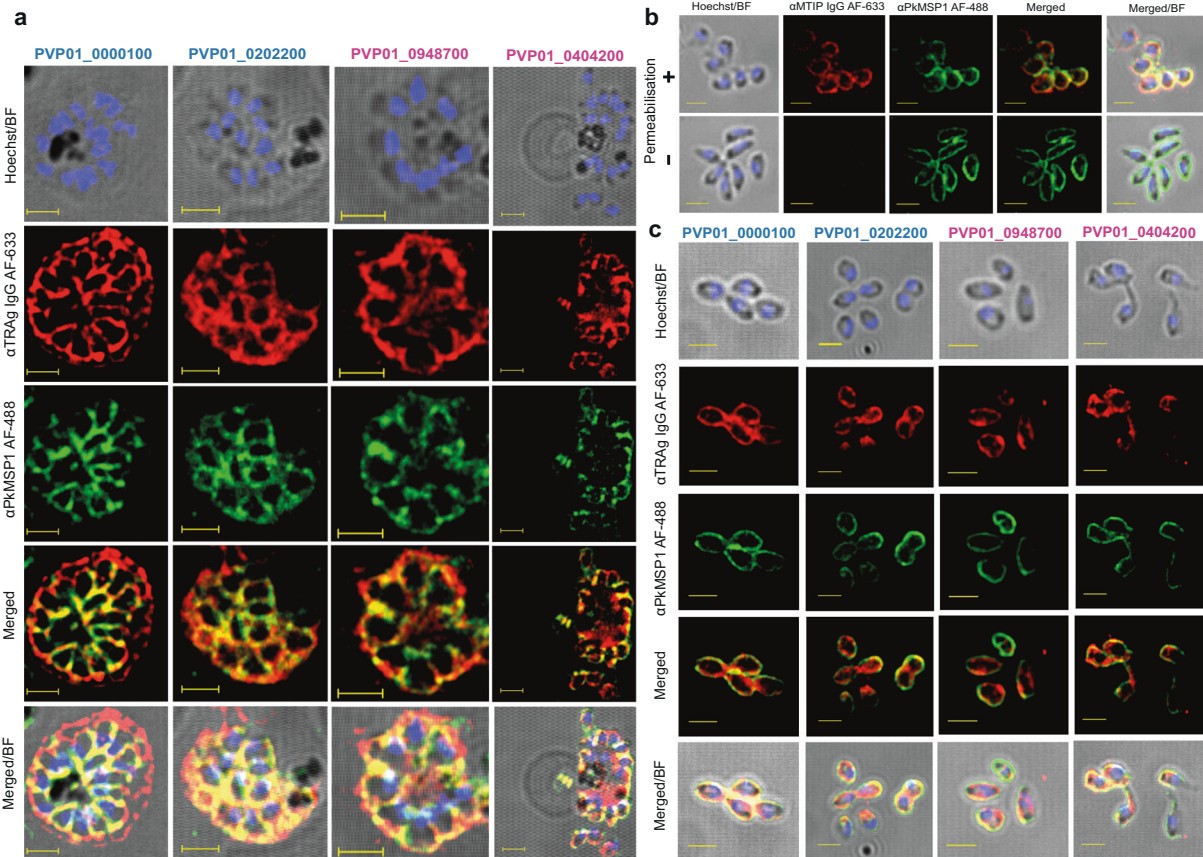

**Fig. 1 | Multiple PvTRAGs localise to the merozoite surface, regardless of signal peptide prediction. a** Rabbit polyclonal IgGs raised against four PvTRAgs (colour coded as Blue for -SP and magenta +SP) were used to study the localisation of the proteins in *P. vivax* isolates. A rat antibody against PkMSP$_{1-19}$ was used as a merozoite surface marker for co-localisation studies. The first column is an overlay of Hoechst with brightfield images, while the last column is an overlay of Alexa Fluor 633, Alexa Fluor 488, Hoechst and brightfield (BF). The highly colocalised fluorescence intensity of the green and red channel indicates the presence of the TRAgs on the surface of the individual Pv merozoites. The colocalised fluorescence intensities are quantified in Supplementary Fig. 4. **b** Triton X100 treated purified *Plasmodium knowlesi* merozoites were stained with anti-*Plasmodium falciparum* MyoA Tail Interacting Protein (MTIP) antibodies, which localise to the Inner Membrane Complex. αMTIP antibody yielded strong peripheral staining following membrane permebalisation (upper panel), but not in the absence of permeablization (lower panel) when the antibodies are unable to penetrate the non permeabilised membrane. Anti PkMSP$_{1-19}$ was used as a merozoite surface marker and the signal remain constant regardless of permeablization as expected. **c** Purified non-permeablilised *P. knowlesi* merozoites were stained with rabbit antibodies generated against four PvTRAgs (± predicted SP); the panel layout is the same as mentioned in Fig. 1a. Each αTRAg rabbit antibody binds to its target on the merozoite outer surface and is colocalised with the merozoite surface marker αPkMSP$_{1-19}$ even in the absence of permeablisation. The Scale bars are two microns and all the images were taken in 100X with AiryScan settings. Alexa Fluor 633 goat-anti rabbit (red) and Alexa Fluor 488 goat-anti rat (green) were used as secondary antibodies for staining. Hoechst 33342 (blue) was used to stain the nucleus. All the micrographs presented are representative of *n* = 3 independent experiments.

background was expected and Pfs25, which includes a Cd4d3 + 4 tag at the C-terminus along with the Biolinker peptide and hexa-His tag was used as a negative control. Only a very low-level of cross-reactivity was found for the anti-PVP01_0000100, PVP01_0532700 and PVP01_0202200 antisera, while anti-PVP01_0404200 had some level of cross-reactivity with one other TRAg, PVP01_0202200 (Supplementary Fig. 2).

Immunofluorescence assays (IFAs) using ex vivo *P. vivax* parasites enriched for schizonts showed merozoite surface localisation for all antisera, whether they were raised against PvTRAgs without (PVP01_0000100 and PVP01_0202200) or with (PVP01_0948700 and PVP01_0404200) a predicted SP. Merozoite surface localisation was defined based on co-localisation with merozoite surface stain marker PkMSP$_{1-19}$ (Fig. 1a), with the Pearson Correlation Coefficient (*r*) or colocalisation averaging 0.8 (Supplementary Fig. 4a). All four anti-PvTRAg antibodies also yielded predominantly merozoite surface localisation in *P. knowlesi*, again irrespective of whether the TRAgs were predicted to contain an SP or not (Supplementary Fig. 3). The *r* value for colocalisation with PkMSP1 was lower in *P. knowlesi*,

presumably because of lower cross-reactivity of anti-PvTRAg antibodies with their *P. knowlesi* orthologues (Supplementary Fig. 4b).

To confirm that staining represented the merozoite surface and not internal organelles such as the inner membrane complex (IMC, a cytoskeleton like structure that sits immediately under the plasma membrane in mature merozoites), IFAs were repeated with purified *P. knowlesi* merozoites using both permeabilising and non-permeabiling conditions. To validate the permeabilisation process the staining was optimised with a control IMC antibody raised against *Plasmodium falciparum* MyoA Tail Interacting Protein (MTIP)[24,25]. As expected the anti-MTIP signal was absent in the non permeabilised merozoites (Fig. 1b, Lower panel), establishing that in the absence of detergent the antibodies were unable to gain access through the merozoite surface membrane to internal organelles, whereas a strong signal for anti-MTIP was observed for the permeabilised merozoites (Fig. 1b, Upper panel). By constrast, non-permeabilised merozoites all stained strongly with antibodies against all four PvTRAgs, as did antibodies against the merozoite surface marker PkMSP$_{1-19}$, confirming that they are all

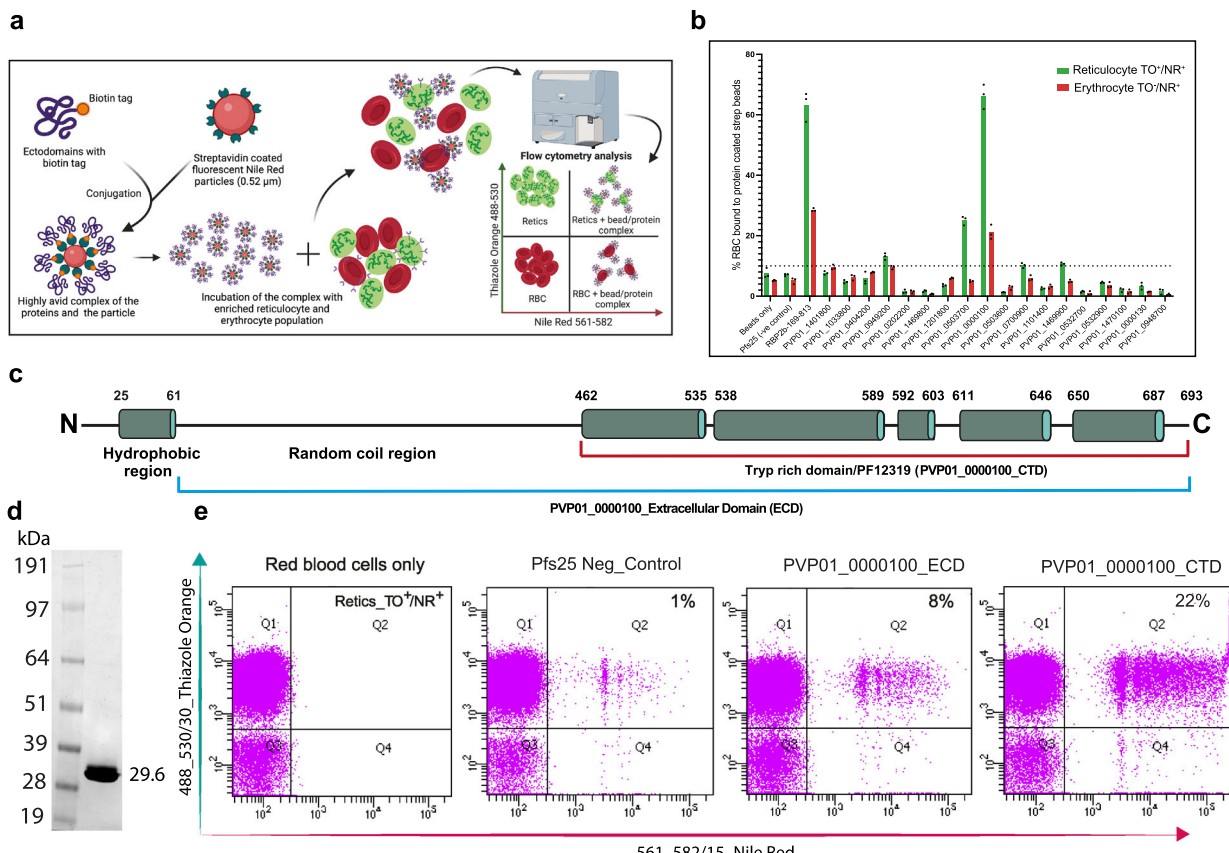

**Fig. 2 | Screening PvTRAgs for erythrocyte and reticulocyte binding activity.**
**a** Affinity purified biotinylated recombinant proteins were multimerised on streptavidin coated Nile Red beads and incubated with a mixture of erythrocytes and reticulocytes. Reticulocytes were labelled with Thiazole Orange (staining rRNA) before binding was assessed using flow cytometry. Erythrocytes are devoid of any genetic material, so are not labelled with TO and are in the lower half of the plots. Any binding with protein-labelled beads will cause a shift in the population of cells towards upper right (reticulocyte, green) or lower right (erythrocyte, red). The figure is created with Biorender.com. **b** Bar graph showing the binding of the multimerised PvTRAgs to reticulocytes and erythrocytes. PvRBP2b$_{169-813}$ previously reported to have binding activity against reticulocytes was used as positive control, while Pfs25, a sexual stage antigen, was used as a negative control for this assay. A cut-off value for positive binding was set as 10% of reticulocytes bound, based on the maximum binding value observed for the bead-only control sample. PVP01_0000100 and PVP01_0503700 are both substantially above this cut-off for both reticulocyte and erythrocyte binding and are termed 'strong binders', while PVP01_0949200, PVP01_0700900 and PVP01_1469900 are above the cut-off for

reticulocytes only, and are considered much weaker. The data represent the average of $n = 3$ technical replicates. The gating strategy and source data are provided in the Supplementary Fig. 7 and Source data file. **c** Schematic diagram of PVP01_0000100 secondary structure prediction using NetSurfP 3.0 and Phobius identifies a long N- terminal random coil region and a C-terminal trp-rich region which is predicted to possess an extensive helical secondary structure. The alpha helices are deep green and the random coil regions are black lines. The ectodomain (ECD) was used in binding assays and antibody generation and the C-terminal domain was used for experimental structure determination and binding assays. The boundaries are indicated for the respective regions. **d** SDS PAGE of purified PVP01_0000100 CTD (amino acids 459-693) following a two-step purification (affinity and size exclusion); protein was stained with Commassie Blue. The SDS page image is a representative of $n = 3$ independent experiment. **e** The reticulocyte binding activity of the PVP01_0000100 CTD and ECD was assessed using flow cytometry as above. All the Retic_TO + /NR + (Q2) populations were calculated after eliminating background population (Q2) from the red blood cells only. This experiment has been repeated $n = 2$ times independently.

localised to the outer surface of the merozoites. Together, these data support the hypothesis that the lack of a predicted SP in at least two PvTRAgs does not preclude them from entering the secretory pathway and becoming localised to the merozoite surface. Given differences in the prediction of SPs in PvTRAgs between servers, and that these servers are often trained primarily on model eukaryotes where amino acid usage can be very different to *Plasmodium* parasites, we hypothesise that the N-terminal hydrophobic region found in all PvTRAgs may actually function as an SP and support their secretion. Whether this hypothesis holds true for all TRAgs requires further validation.

**PVP01_0000100 binds preferentially to human reticulocytes via its C-terminal domain**
Given the localisation of multiple TRAgs to the merozoite surface (Fig. 1b and refs. [14,26]) we hypothesised that they may bind directly to

the surface of erythrocytes or reticulocytes and play a role in parasite invasion. To test this, the ECDs (as defined above) of all 33 PvTRAgs were trialled for expression using the HEK293E system. The expressed proteins are secreted into the media with a C-terminal hexa-His tag and a 31-amino acid biotinylation peptide, which can be used for affinity purification and to detect expression. Expression of 27 of the 33 PvTRAgs was confirmed by Western blot using HRP conjugated anti-His antibody (Supplementary Fig. 5). Sufficient yields for binding studies were obtained for 15 PvTRAgs. To increase avidity, affinity purified PvTRAg proteins were multimerised on Nile Red (NR) coated streptavidin beads (Fig. 2a). Reticulocytes enriched using Percoll cushions followed by anti-CD71 magnetic beads (Supplementary Fig.6) were mixed 1:1 with mature erythrocytes and then incubated with the PvTRAg-NR beads. After incubation and washing, the mixture was stained with Thiazole Orange (TO), which binds the RNA present in reticulocytes but not mature erythrocytes and allows

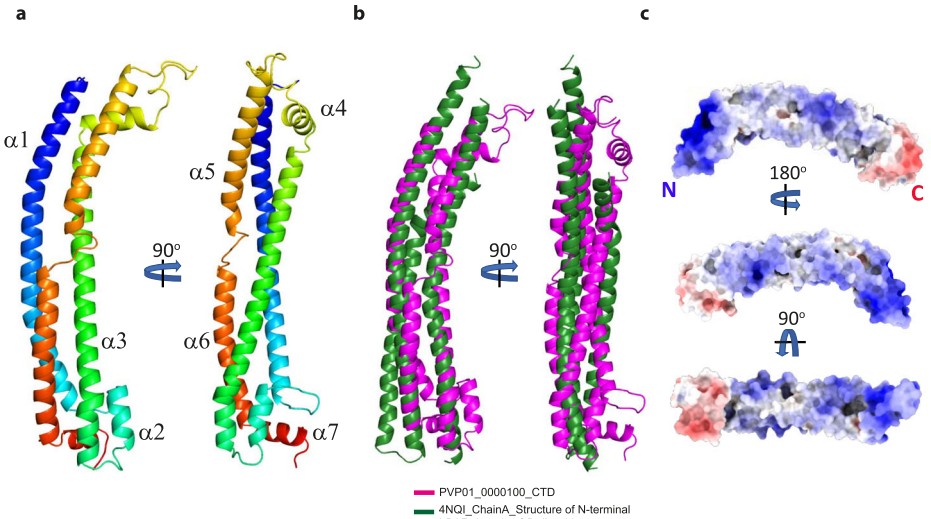

PVP01_0000100_CTD

4NQI_ChainA_Structure of N-terminal
I-BAR domain of *D.discoideum*

**Fig. 3 | Experimentally determined structure of the PVP01_0000100 CTD and its structural homology with BAR domain proteins. a** Ribbon diagram of the experimentally determined structure of the CTD of PVP01_0000100 coloured from blue (N-terminus) to red (C-terminus). Two orientations are displayed (rotated by 90°) demonstrating the extended, slightly bent, three-helical bundle structure. **b** Superposition of the CTD of PVP01_0000100 (magenta) with the BAR domain from *Dictyostelium discoideum* (PDB ID 4NQI, green) showing the similarity between the fold of these two domains. **c** Electrostatic potential analysis of the surface of PVP01_0000100 identifies positively charged patches at the N-terminal end and along the surface including the concave side of the protein. The small domain at the C-terminal end of the TRAg domain possesses a negatively charged surface.

detection of binding to reticulocytes and/or erythrocytes by flow cytometry, double gating with NR and TO. Pfs25, a sexual stage protein from *P. falciparum*[27–29] with no known red cell binding activity was used as a negative control, while the reticulocyte-specific CD71-interacting domain of PvRBP2b$_{169-813}$[8] acted as a positive control. The cut-off for positive binding was established at 10% of reticulocytes bound, which corresponds to the highest value observed for the bead-only control sample and is nearly equivalent to the binding level of the negative control protein (Pfs25) plus six times the standard deviation of Pfs25 binding; proteins displaying binding above this cut-off were considered significant. Of the 15 PvTRAgs tested in binding assays, PVP01_0000100 (>60%) and PVP01_0503700 (>20%) were the most significant, and both showed a clear preference for reticulocytes over erythrocytes (Fig. 2b), similar to the preference of PvRBP2b. (Fig. 2b; gating strategy shown in the Supplementary Fig. 7). PVP01_0949200, PVP01_0700900 and PVP01_1469900 also showed binding above the cut-off but were considered to be much weaker binders.

PVP01_0000100 and PVP01_0503700 are close to each other in the TRAg phylogenetic tree (Supplementary Fig. 1) and share 34% sequence identity across the full-length proteins, and 48% identity in their C-terminal Tryptophan/Threonine-rich domains. Secondary structure prediction for PVP01_0000100 using NetSurfP 3.0 suggests that the Tryptophan/Threonine-rich domain possesses a high α-helical content (Fig. 2c). To test which domain was responsible for reticulocyte binding, the C-terminal region (amino acids D459–693L) was expressed alone (Fig. 2d) and assessed alongside the ECD (amino acids K62–693L). The C-terminal Tryptophan/Threonine-rich domain (CTD) showed a higher level of binding towards reticulocytes than the ECD (Fig. 2e), strongly suggesting that reticulocyte binding activity is mediated by this domain.

To control for potential interactions between the TRAg-coated Nile Red beads and the anti-CD71 beads used to enrich for reticulocytes, the assay was also performed with reticulocytes enriched using Percoll alone. Both PVP01_0000100 and the positive control PvRBP2b bound to Percoll-enriched reticulocytes, while Pfs25 did not, establishing that binding was not impacted by the method of reticulocyte purification (Supplementary Fig. 8).

## The PVP01_0000100 C-terminal Tryptophan/Threonine-rich domain adopts a helical fold with structural homology to Bin/Amphiphysin/Rvs (BAR) domains

To establish how the C-terminal Tryptophan/Threonine-rich domain of PVP01_0000100 mediates reticulocyte binding we used X-ray crystallography to experimentally determine the structure of this domain. The structure was refined to 1.45 Å resolution and had one molecule in the asymmetric unit (Fig. 3a and Supplementary Table 4). Consistent with the NetSurfP predictions, the structure is composed primarily of α-helices that are arranged into an extended three-helical bundle with small helical domains at each end. The conserved tryptophan residues that define the domain are distributed along the length of the helical bundle and are all buried within the fold, contributing to the arrangement of the long helices. Since experimental determination of this structure, new methods for prediction of 3D protein structures have been accelerated through use of deep-learning strategies as exemplified by AlphaFold2 (AF2)[30]. AF2 predictions of the PVP01_0000100 Tryptophan/Threonine-rich domain structure closely resemble the structure we experimentally determined, with an RMSD of 1.56 Å over 213 Cα atoms (Supplementary Fig. 9a).

The accuracy of the predicted AF2 structure therefore allows for comparison of our experimental structure for PVP01_0000100 with AF2 predictions for the Tryptophan/Threonine-rich domain of TRAgs from other *Plasmodium* species. The quality of the predicted models was assessed by analysis of the pLDDT plots which showed high confidence scores for the helical regions of these predicted models (Supplementary Fig. 9c). Superposition of PVP01_0000100 Tryptophan/Threonine-rich domain with its closest orthologue in *P. knowlesi* (PKNH_1300500, RMSD 1.83 Å over 211 Cα atoms, sequence identity 77.4%) and a much more distant orthologue in *P. falciparum* (PF3D7_0102700, RMSD 2.87 Å over 179 Cα atoms, sequence identity 30.2%) suggests strongly that the three-helical fold we experimentally determined is highly conserved across *Plasmodium* species (Supplementary Fig. 9b). To compare the conservation of the fold we also generated AF2 models for multiple paralogues in *P. vivax*. The C-terminal domains of four PvTRAgs were selected across the phylogenetic tree which range in sequence identity relative to PVP01_0000100 CTD from 28 to 42%. Superposing the Cα atoms of

AF2 models onto our experimentally confirmed PVP01_0000100 CTD structure indicate a high level of conservation, with RMSD values ranging from 1.2 to 2 Å (Supplementary Fig. 10a). In addition, we compared the TRAg domain of PVP01_0000100 with AF2 predictions of the TRAg domains from seven different species of *Plasmodium* (*P. falciparum, P. malariae, P. berghei, P. yoelii, P. ovale, P. cynomolgi* and *P. knowlesi;* in each case we selected the TRAg within the species that had the lowest primary sequence identity to PVP01_0000100). The RMSD values strongly support that the overall fold of TRAg domains are highly conserved across the entire *Plasmodium* genus, with the main differences being within short helices and loops at the distal ends of the extended helical core (Supplementary Fig. 10b).

The structure of the Tryptophan/Threonine-rich domain of PVP01_0000100 was compared to other known protein folds using DALI[31]. This identified structural homology to several membrane-binding proteins containing BAR (Bin/Amphiphysin/Rvs) domains (PDB ID 4NQI with a DALI Z-score of 7.2) (Fig. 3b). BAR domains possess extended helical bundles with a bent shape that can bind lipids and in some cases reshape membranes. Typically, BAR domains interact with negatively charged lipids on the plasma membrane via positively charged patches on the protein surface[32–35]. Analysis of the surface charge of the PVP01_0000100 revealed extended positively charged patches, mainly on the N- terminal and concave side of the protein surface (Fig. 3c), similar to the shape and surface charge of BAR domains, suggesting a potential lipid binding function.

## The PVP01_0000100 Tryptophan/Threonine-rich domain binds to sulfatide

In order to carry out comparative in vitro lipid-binding assays, the ECD and CTD of PVP01_0000100, and a negative control protein, Pfs25, were all purified as above for RBC binding assays (Fig. 4a). As PVP01_0000100 is expressed on the merozoite surface, the lipid species it is likely to interact with will be those present in the outer leaflet of the host plasma membrane. We therefore performed lipid-binding assays with binding strips containing a range of sphingolipids, which are present in the outer leaflet of cell surface membranes. PVP01_0000100 CTD showed specific and strong binding to sulfatide (Fig. 4b), an anionic glycosphingolipid found on the outer leaflet of multiple cell types, including red blood cells[36,37]. Commercial sphingostrips are made by dissolving lipids in organic solvent and spotting them onto a nitrocellulose membrane, resulting in lipid head groups being randomly arrayed rather than presented in a specific orientation as they would be in a membrane. These strips therefore provide a useful rapid screening tool to test binding to a wide range of lipids, but do not necessarily recapitulate physiological lipid binding. To test binding in a more relevant context, the interaction with sulfatide was confirmed using giant multilamellar liposomes composed of 48% phosphatidylcholine (PC), 2% Rhodamine-labelled phosphatidylethanolamine (Rhod-PE) and 50% sulfatide. These liposomes were incubated with 1 μM purified protein, pelleted via centrifugation, washed, resolved using SDS-PAGE and stained with Coomassie. Both PVP01_0000100 ECD and CTD bound more strongly to liposomes containing 50% sulfatide than they did to those without (Fig. 4c). Pfs25, the negative control, did not appreciably bind either liposome composition.

These two independent assays confirm the lipid-binding properties of the PVP01_0000100 Tryptophan/Threonine-rich domain that were suggested by the structural homology to BAR domains. To explore the specificity of lipid binding and specifically the influence of charge, liposome assays were repeated with liposomes consisting of phosphatidylserine (PS), another negatively charged lipid but with a very different headgroup. A titration experiment was performed with liposomes composed of increasing concentration (0–80%) of sulfatide or PS and binding quantitated using densitometry (Fig. 4d). PVP01_0000100 binding to sulfatide followed a sigmoidal binding curve, suggesting a saturable and hence specific interaction. The sigmoidal binding relationship for the interaction between sulfatide and PVP01_0000100 also suggests there are multiple cooperative sulfatide binding sites. Binding to PS-containing liposomes did not follow this pattern, instead the binding appeared linear up to 50% PS and then decreased at higher percentages, suggesting the interaction is less specific. Therefore, lipid headgroup charge alone does not appear to be the sole mediator of PVP01_0000100 binding to sulfatide.

Given the predicted structural conservation of the core *Plasmodium* Tryptophan/Threonine-rich domain fold (Fig. 3b, Supplementary Fig. 9), we hypothesised that lipid-binding may be a conserved feature of this domain in multiple *Plasmodium* TRAgs. To test whether other PvTRAgs also have lipid-binding properties, five were chosen based on their expression level in HEK293E cells and to represent a range of primary amino acid identity to PVP01_0000100 (PVP01_0202200-31%, PVP01_0404200-29%, PVP01_0948700-19%, PVP01_1401800-30%). The ECDs of this small subset of PvTRAgs were purified using affinity and size exclusion chromatography (Supplementary Fig. 11a) and assayed for lipid binding using liposome binding assays. All four PvTRAgs had lipid-binding activity, but with different preferences (Supplementary Fig. 11b). Similarly to the PVP01_0000100 ECD, the PVP01_0948700 ECD showed a strong preference specifically towards sulfatide-containing liposomes compared to those containing only PC or both PC and PS. By contrast, PVP01_0202200, PVP01_0404200 and PVP01_1401800 all showed some level of binding to all three liposome compositions but with an apparent weaker affinity. This result supports the hypothesis that multiple PvTRAgs bind to membranes and that the lipid headgroup preference differs. Neurofascin 155 (NF155), a known sulfatide binding protein[38], was used as the positive control for this assay (Supplementary Fig. 11b).

## The *P. knowlesi* homologue of PVP01_0000100 plays a role in reticulocyte invasion

*P. vivax* parasites cannot be cultured in vitro, which means the function of PVP01_0000100 cannot be directly tested in this species. To test the implications of sulfatide binding we therefore generated a knockout line of the *PVP01_0000100* orthologue (*PKNH_1300500*) in *P. knowlesi*, a close phylogenetic relative of *P. vivax* which can be cultured in human erythrocytes[39]. Purified PKNH_1300500 bound both sulfatide and reticulocytes, like PVP01_0000100 (Supplementary Fig. 12). Deletion of PKNH_1300500 was carried out using a CRISPR-Cas9 two-plasmid approach, resulting in deletion of the entire gene and replacement with a GFP expression cassette (Fig. 5a; see Methods for details). After transfection and selection, genotyping and GFP fluorescence revealed a mixed population of wildtype and edited parasites, from which knockout lines were isolated by dilution cloning (Fig. 5b). We have previously shown that anti-PVP01_0000100 antiserum detects merozoite surface staining in wild type *P. knowlesi* schizonts (Supplementary Fig. 3). The fluorescence intensity of this staining was reduced by >80% in the *PKNH_1300500* knockout line (Fig. 5c), confirming gene deletion. The weak residual staining suggests that there might be some low-level cross-reaction of anti-PVP01_0000100 antisera with PkTRAgs other than its closest homologue PKNH_1300500, although previous tests showed only limited cross-reaction (Supplementary Fig. 2).

To assess the impact of *PKNH_1300500* deletion, growth assays were performed in duplicate for 14 days (c. 14 cycles, as *P. knowlesi* has a 27-h cycle). A slight reduction in cumulative growth of the *PKNH_1300500* KO line compared to wildtype *P. knowlesi* was observed (Fig. 5d). As both PVP01_0000100 and PKNH_1300500 bind preferentially to reticulocytes, we carried out invasion assays comparing the ability of the KO line to invade both reticulocytes and mature erythrocytes. Late stage parasites were incubated with a mixture of erythrocytes and reticulocytes and invasion was quantitated using flow cytometry, with anti-CD71 staining distinguishing between

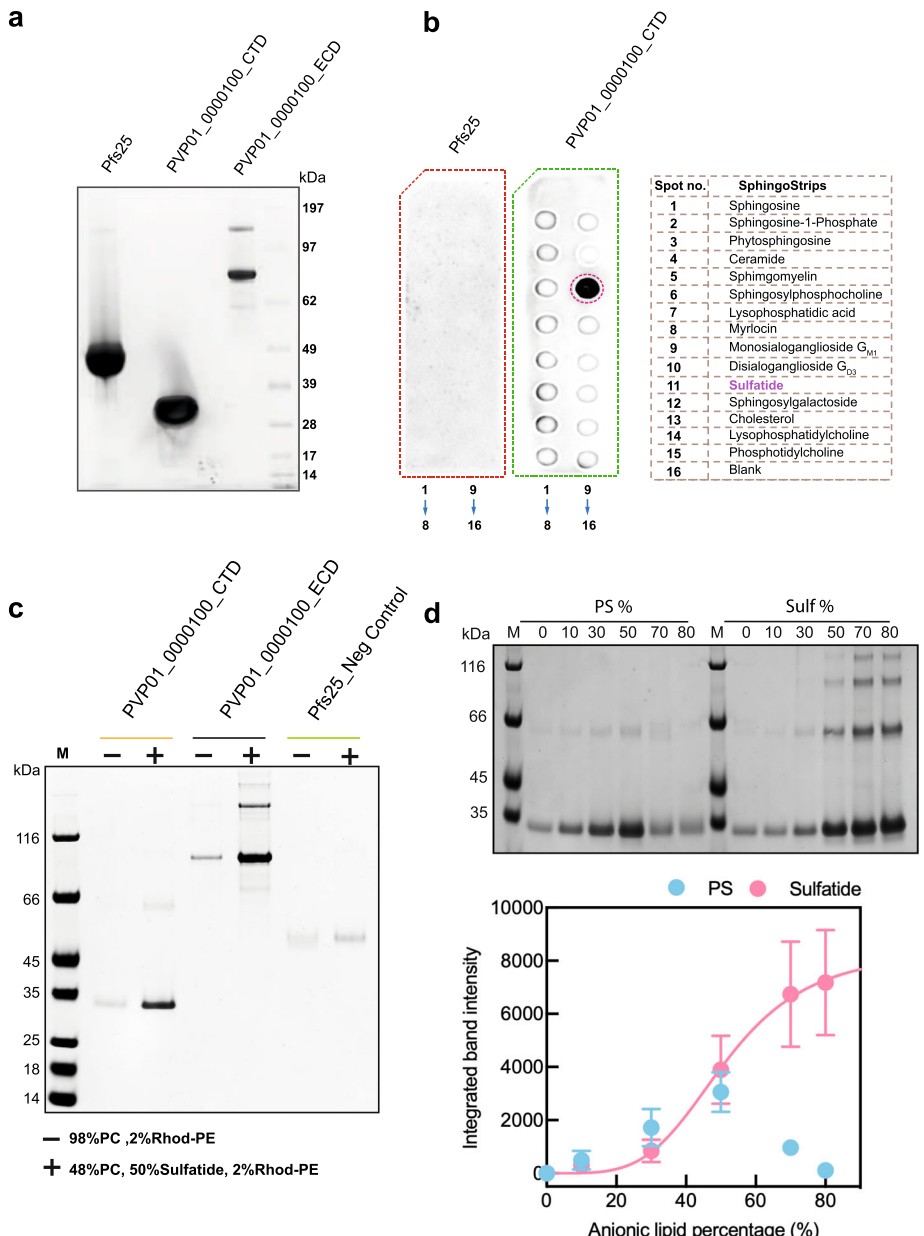

**Fig. 4 | Lipid binding properties of PVP01_0000100. a** SDS-PAGE of purified PVP01_0000100 ECD, PVP01_0000100 CTD and Pfs25 following affinity and size exclusion chromatography. The SDS page image is a representative of $n = 3$ independent experiment. **b** PVP01_0000100 CTD and Pfs25 (1 μg) were incubated with sphingostrips and binding detected using HRP conjugated anti-His antibody (1:5000) and chemiluminescent substrate. The spot numbers and orientation are shown under the blot, and show strong binding of PVP01_0000100 CTD to sulfatide (spot no.11, magenta circle). The lipid dot blot is representative of $n = 3$ independent experiments. **c** Liposomes composed of PC:Rhod·PE only (−, 98:2) or PC:sulfatide:Rhod·PE (+, 48:50:2) were incubated with 1 μM PVP01_0000100 ECD, CTD or Pfs25 and then run on SDS-PAGE and stained with Coomassie. The band intensity is increased in the presence of sulfatide for PVP01_0000100 CTD and ECD,

while the negative control, Pfs25, did not significantly bind liposomes. The SDS page image is representative of $n = 3$ independent experiments **d** Liposomes containing increasing percentages of sulfatide and another anionic lipid phosphatidylserine (PS) (0-80%) were incubated with 1 μM PVP01_0000100 CTD before being run on SDS-PAGE and stained with Coomassie. The additional bands above the CTD are higher order oligomers produced as a by-product of sample preparation prior to SDS-PAGE. Assay performed in (**d**) was done in triplicate and quantified using ImageJ and GelAnalyzer. The integrated band intensity, proportional to the amount of protein bound, was plotted against lipid percentage (average ± S.E.M., $n = 3$ independent experiments) and revealed a sigmoidal binding relationship for sulfatide and a linear binding relationship for PS up to 50%, followed by a dramatic decrease in binding at higher percentages.

erythrocytes and reticulocytes, and invasion events quantitated using SYBR Green to label parasite DNA (Supplementary Fig. 13). Using a CD71 positive cutoff to define reticulocytes carries the risk of false negatives as older reticulocytes may have very low CD71 levels[40], but it reduces the risk of false positives from low levels of contaminating erythrocytes that would also be CD71 negative. Wildtype *P. knowlesi*, while not restricted to reticulocytes like *P. vivax*, does have a clear preference for these immature red cells[41]. The preference was

observed as expected in wildtype *P. knowlesi*, whereas in *PKNH_1300500* knockout parasites the preference was significantly reduced (Fig. 5e). The impact of *PKNH_1300500* deletion specifically on reticulocyte invasion likely explains the lack of difference in cumulative growth rate between KO and wildtype lines, as under standard growth conditions, parasites are cultured only in the presence of mature erythrocytes, with few if any reticulocytes present (as these mature rapidly in culture).

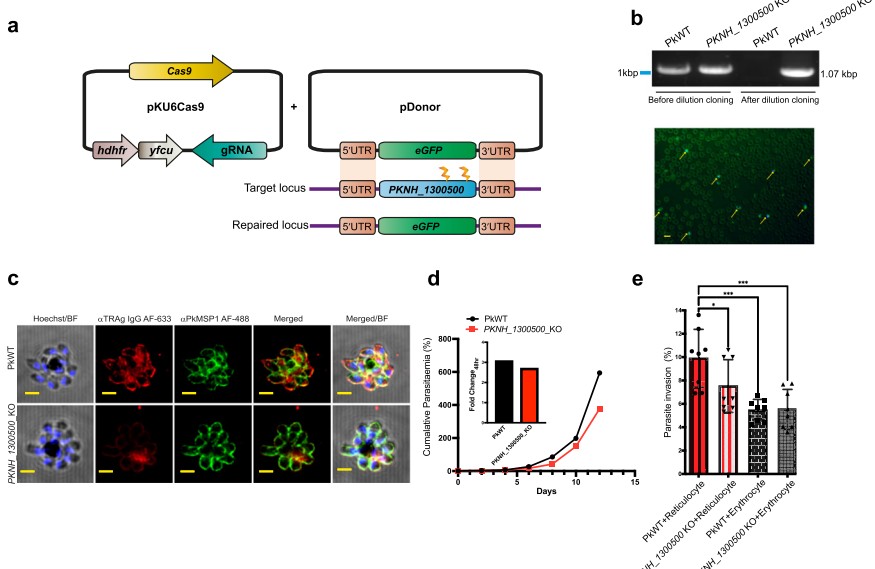

**Fig. 5 | Gene deletion in *P. knowlesi* reveals a function for *PKNH_1300500* in reticulocyte invasion. a** The homologous repair template used for gene editing of the *PVP01_0000100* orthologue in *P. knowlesi* (*PKNH_1300500*) contained an eGFP expression cassette flanked by 5' and 3' UTR 800 bp homology regions. This construct was co-transfected with a plasmid expressing both Cas9 and a guide RNA. Two guide RNAs were used that are positioned near the 3' end of the gene. **b** After transfection and drug (100 nM pyrimethamine) selection, genotyping PCR revealed a mixed population of PkWT and *PKNH_1300500* KO parasites, as indicated by PCR specific for the wildtype (WT) and knockout (KO) locus. A pure population of the edited line was obtained by dilution cloning, with no WT genotyping visible (lane 3), (uncropped, unprocessed image is provided in the source data file). Successful editing was also assessed using GFP expression (yellow arrows showing the positively edited cells) using fluorescence microscopy, *n* = 3 independent experiment. Scale bar = 10 µm. **c** Rabbit polyclonal IgG raised against PVP01_0000100 was used to detect its orthologue PKNH_1300500 in *P. knowlesi*. Top panel is the *P. knowlesi* wild type population and bottom panel is *PKNH_1300500* gene edited population.

PkMSP$_{1-19}$ antibody (raised in rat) was used as merozoite surface marker and Hoechst was used to stain the parasite nucleus. The images were taken in 100X with AiryScan settings and are repsentative of *n* = 2 independent experiments. Scale bars are two microns. **d** Growth rate of the PkWT and *PKNH_1300500* KO line was assessed for 14 days, with parasites split every 2 days and reset to 1% parasitemia. Cumulative parasitaemia was calculated by multiplying parasitemia values detected with flow cytometry with the cumulative split values. Data points represent two independent biological replicate. **e** Invasion was quantitated using flow cytometry, with parasites detected using SYBR green staining (Q1), and erythrocytes and reticulocytes distinguished using anti-CD71. The percentage of reticulocytes (Q2) or erythrocytes (Q1) that were invaded is shown (mean ± S.E.M) and the data combines three biological replicates, one containing technical duplicate wells, and the other two technical triplicate wells. $P^* = 0.0207$, $P^{***} = 0.0003$ (One-way ANOVA with Sidak's multiple comparisons test was used to calculate *p* values). The gating strategy is shown in Supplementary Fig. 13 and source data is provided in the source data file.

To explore the mechanism by which PVP01_0000100 might play a role in invasion, we incubated purified PVP01_0000100 CTD with reticulocytes and assessed the impact on reticulocyte biophysical properties using flickering spectroscopy[42]. The PVP01_0000100 CTD led to a significant decrease in membrane tension, which in *P. falciparum* has previously been shown to be directly linked with invasion efficiency, with lower tension erythrocytes being more accessible for invasion than higher tension erythrocytes[43]. Purified Pfs25 used at the same concentration had no effect (Fig. 6a). Tension is thought to affect invasion through its impact on the membrane wrapping energy, essentially the amount of energy required for the parasite to wrap the cell membrane around itself during the invasion process[44]. Addition of PVP01_0000100 reduced the wrapping energy compared to untreated reticulocytes (Fig. 6b), suggesting a model by which PVP01_0000100 binding to sulfatide modulates invasion by decreasing the energy required for merozoites to enter their host.

## Discussion

TRAgs have been studied in multiple *Plasmodium* species and are potential vaccine targets. However, a clearly defined molecular function for these antigens has not yet been established and some of the published literature is contradictory, suggesting a thorough systematic analysis is required. A comprehensive search and annotation of TRAgs from *P. vivax, P. cynomolgi* and *P. knowlesi* revealed that half of the genes are predicted to contain signal peptides, but the other half are not. From a cell biology and evolutionary perspective, it would be unusual for the same protein

family to contain members both with and without signal peptides, as secreted and cytosolic proteins generally have distinct functions. A more prosaic, technical explanation therefore seems likely. The genomes of *Plasmodium* species are all biased towards a high AT content and have patterns of codon usage that are unusual compared to other eukaryotes[45,46]. This can affect the efficiency of automated genome annotation for protein features such as signal peptides, which use domain prediction programmes that are usually trained on genomes from humans or model eukaryotes. We observed several TRAgs had SPs annotated by some prediction programmes but not others, and all contained a hydrophobic sequence at their extreme N-terminus, regardless of SP annotation. Immunofluorescence assays using antibodies raised against four *P. vivax* TRAgs, two with predicted SPs and two without, revealed that all four co-localised with a merozoite surface marker, indicating that all enter the secretory pathway. Based on this, we hypothesise that all PvTRAgs are potentially secreted, regardless of whether the hydrophobic region at their N-terminus is formally annotated as a signal peptide; formal proof of this clearly requires testing of further PvTRAgs. This also emphasises the limitations that automated genome annotation can sometimes have in inferring gene function, and the importance of researchers validating annotation and exploring the use of multiple annotation servers when analysing *Plasmodium* genes, particularly those of unknown function.

The presence of multiple PvTRAgs on the merozoite surface, confirmed by this and previous studies[26], suggests a role for some

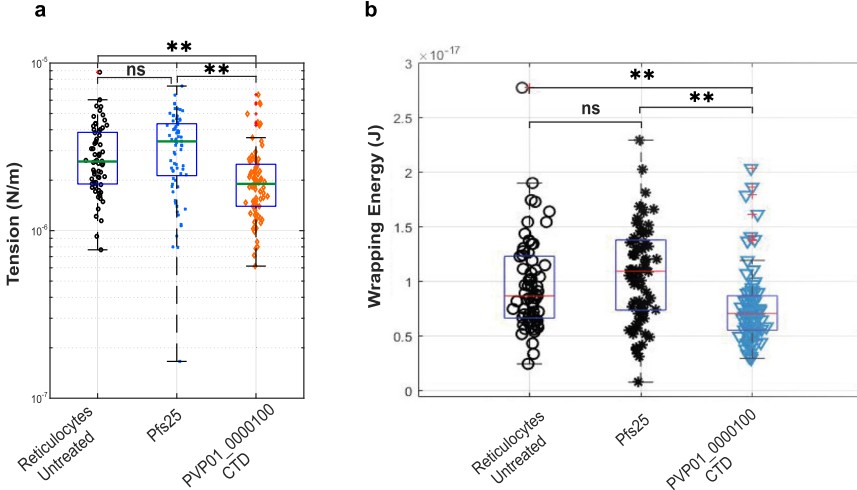

**Fig. 6 | Impact of PVP01_0000100 binding on reticulocyte membrane properties. a** Membrane tension was measured using flickering spectroscopy for CD71+ reticulocytes in the absence of additional protein (untreated), and incubated with 100 µg/ml proteins (Pfs25 and PVP01_0000100 CTD). Number of cells per sample: untreated $n = 65$, Pfs25 $n = 73$, and PVP01_0000100 $n = 79$. Pairwise comparisons between groups used Tukey's honest significant difference (HSD) test and Bonferroni correction. $P^{**} = 0.0015$ (Reticulocytes Untreated vs PVP01_0000100 CTD), $P^{**} = 1.7 \times 10^{-6}$ (Pfs25 vs PVP01_0000100 CTD). The fluctuations with contour detection of untreated red blood cell and in presence of proteins are shown in Supplementray Movies 1,2 and 3. **b** Wrapping energy was calculated using MATLAB.

Data points for untreated are 57, for Pfs25 are 73 and for PVP01_0000100 sample are 77. Statistical comparison across groups was performed by one-way ANOVA with Bonferroni correction for multiple comparisons, while pairwise comparisons between groups used the Tukey HSD test. $P^{**} = 5.2 \times 10^{-4}$ (Reticulocytes Untreated vs PVP01_0000100 CTD), $P^{**} = 1.3551 \times 10^{-5}$ (Pfs25 vs PVP01_0000100 CTD). Each box, the central mark indicates the median, and the bottom and top edges of the box indicate the 25th and 75th percentiles, respectively. The whiskers extend to the most extreme data points not considered outliers, and the outliers are plotted individually using the '+' marker symbol. Source data is provided in the source data file.

family members in erythrocyte and reticulocyte binding. Using a new bead binding assay we observed strong binding for PVP01_0000100 and PVP01_0503700 with a preference for reticulocytes in both cases, which has not been reported previously in studies of other PvTRAgs[12,13]. PKNH_1300500, the *P. knowlesi* homologue of PVP01_0000100, also showed reticulocyte binding activity, although this was weaker than PVP01_0000100, potentially due to lower protein expression affecting the bead conjugation efficiency. Experimental determination of the crystal structure of the Tryptophan/Threonine-rich domain of PVP01_0000100 suggested a hypothesis for the molecular mechanism of this binding activity. The three helical bundle structure of this domain has homology with the BAR domain superfamily, which is characterised by α-helical coiled coils that form a crescent shape with positively charged patches on the concave side, which interact with phospholipid membranes[32,47]. BAR domains are often involved in detecting or modulating membrane curvature, for example by interacting with negatively charged membranes and contributing to membrane bending during endocytosis[48], and can also serve as membrane-binding scaffolds for other proteins, thereby participating in signalling pathways[49]. The PVP01_0000100 Tryptophan/Threonine-rich domain has positively charged patches on the concave side (driven by high lysine content, 18%), and lipid dot blot and liposome binding assays confirmed that both it and its orthologue in *P. knowlesi* (PKNH_1300500) specifically binds to the anionic lipid sulfatide. Comparison of PVP01_0000100 TRAg binding to liposomes that maintain a similar charge but different headgroup (PS versus sulfatide) suggest the membrane binding of PVP01_0000100 possesses specificity for sulfatide. Furthermore, the sigmoidal relationship between sulfatide percentage and TRAg binding suggests the TRAg domain may possess multiple co-operative lipid binding sites. Although several well-characterised BAR domain-containing proteins primarily interact with membranes via relatively non-specific electrostatic interactions this is not the case across the diverse BAR-domain containing protein family. Several BAR domains have been shown to possess specific

differences in their lipid binding properties[50,51] consistent with the binding to sulfatide seen here.

Sulfatides (also known as 3-O-sulfogalactosylceramides, sulfated galactocerebrosides, or SM4) are sphingolipids that participate in a wide range of cellular processes including protein trafficking, cell adhesion and aggregation, axon-myelin interactions, neural plasticity, and immune responses, among others[52]. They are particularly enriched in the myelin sheath surrounding nerve cells but are also found in the extracellular leaflet of the plasma membrane of most eukaryotic cells including red blood cells[53]. Sulfatides have previously been reported to play a role in malaria infection, specifically during the invasion of hepatocytes where circumsporozoite protein has been reported to bind to sulfatide[54,55]. Sulfatide binding has not previously been implicated in erythrocyte invasion in any *Plasmodium* species, but deletion of *PKNH_1300500*, which binds sulfatide like its *P. vivax* orthologue, affected parasite invasion of reticulocytes, suggesting a direct link between sulfatide binding and invasion. The structural homology between the PVP01_0000100 Tryptophan/Threonine-rich domain fold and the BAR domain suggests a clear hypothesis for how TRAgs may promote invasion. TRAg binding to lipids on the surface of erythrocytes could lead to membrane remodelling, potentially affecting biophysical properties and aiding the invasion process, which requires significant reshaping of the erythrocyte membrane in order for the merozoite to enter. In keeping with this hypothesis, addition of purified PVP01_0000100 to reticulocytes reduced their surface tension (Fig. 6a). Lower surface tension has recently been associated with increased invasion efficiency in *P. falciparum*[43], supporting the hypothesis that TRAg-lipid interactions could directly promote invasion of reticulocytes. This interaction is not absolutely essential for invasion, as PKNH_1300500 knockout parasites can still invade both erythrocytes and reticulocytes, albeit at a reduced rate in reticulocytes. This is likely due to the large number of genes within the TRAg family, with other PkTRAgs potentially binding to either sulfatide or other lipids, providing partially overlapping functions. Whether the sulfatide

binding preference of PVP01_0000100 also explains its reticulocyte binding preference is not clear, as we were unable to find any data quantifying the abundance of sulfatide on reticulocytes compared to erythrocytes. It is worth noting though that CD71+ reticulocytes have a c. 30% larger surface area compared to mature erythrocytes, and membrane lipids vesiculate out during the process of RBC maturation, suggesting that many lipids, including sulfatide, are likely to be different between reticulocytes and mature erythrocytes[40,56].

We have used the localisation and structure of a specific merozoite surface localised PvTRAg, PVP01_0000100, to generate and test a specific hypothesis that this protein binds lipids and thereby promotes the invasion of red blood cells by *P. vivax* merozoites. An important question is whether this hypothesis holds true across all TRAgs within *P. vivax*, or indeed for all TRAgs across the *Plasmodium* genus. AlphaFold2 predicts that the structure we experimentally determined for the PVP01_0000100 Tryptophan/Threonine-rich domain is likely to be conserved across *Plasmodium* TRAgs, despite primary sequence conservation levels <50%. This strongly suggests that all will possess a similar extended, helical fold with homology to BAR domains. This conserved fold supports the hypothesis that all TRAgs may possess lipid-binding properties. Using five different PvTRAgs we demonstrate that all can bind liposomes but with some differences in lipid-binding specificities. Given the low primary sequence homology and different surface charge patterns seen between different *Plasmodium* Tryptophan/Threonine-rich domain models, it is very possible that individual PvTRAgs may bind to different lipids, and initial tests of other PvTRAgs support that they have differences in binding activity to sulfatide. However, future studies are clearly required to explore the extent and specificity of lipid-binding across the C-terminal domain of TRAgs from different *Plasmodium* species. Other non-lipid binding functions for individual TRAgs have also been proposed, including binding to the most abundant RBC surface protein, Band 3[13]. While the evidence for this interaction currently lacks structural or quantitative confirmation, it is not necessarily inconsistent with the data presented here. The focus of this work is on the structurally conserved C-terminal domain, and it is quite possible that the more variable N-terminal domains of PvTRAgs could encode other functions such as specific receptor binding, including to Band 3.

It is also worth noting that it has been suggested that other *P. vivax* TRAgs may be exported into the infected erythrocyte cytosol, where they associate with Maurer's Clefts or other membrane structures[57]. Similarly, a *P. berghei* TRAg has recently been shown to be found inside infected erythrocytes and proposed to be involved in membrane remodelling[58]. These observations could still be consistent with the hypothesis that many TRAgs may have lipid binding properties, but with potentially different lipid specificities and in different intracellular locations. It is also possible that TRAg function could be different in *P. berghei* and *P. falciparum*, which only possess a small number of TRAgs, likely representing the ancestral state, compared to *Plasmodium* subgenus species such as *P. vivax* where the family has expanded 5–10-fold, and which from phylogenetic analysis appears to be still expanding within species. Importantly, this work has established a defined molecular function for the TRAg domain and provides clear hypotheses to test across *species* in the further exploration of this unique *Plasmodium* protein family.

## Methods

### Phylogenetic tree generation
The amino acid sequences of TRAgs from *P. vivax* (38), *P. cynomolgi* (33), *P. knowlesi* (29) were obtained from PlasmoDB[20,59]. The full length (including the predicted signal peptide /N-terminal domain) amino acid sequences of these TRAgs were aligned using Clustal Omega[60] and trees generated using PhyML[61]. Phylogenetic trees were constructed

using maximum likelihood method[62]. The evolutionary distances were computed using Jones-Taylor-Thornton Matrix model with a bootstrap test of 100 replicates[63] and remove gap from alignment option.

### Gene synthesis and protein expression
The region corresponding to the entire extracellular domain of 33 *P. vivax* Tryptophan rich antigens were determined using signal peptide[64] and transmembrane[65] prediction servers. Five of the 38 total *P. vivax* TRAgs were omitted because they were >200 kDa in size; proteins above this size are usually not expressed well in HEK239E cells[23]. To prevent the aberrant glycosylation of the proteins in mammalian cells all potential N-linked glycosylation sites (N-X-S/T/P), which are non-functional in *Plasmodium* parasites, were mutated by substituting alanine for serine/threonine/proline. The designed constructs were synthesised (TWIST biosciences, UK) with codons optimised based on the human genome. The gene region encoding the entire extracellular domain was flanked by was flanked by NotI and AscI sites and subcloned using these enzymes into a mammalian expression plasmid that contained a 5' mouse variable κ light chain signal peptide and an enzymatic biotinylation sequence/hexa-His tag[66] at the C-terminus, just as we have previously done for multiple proteins in both *P. falciparum* and *P. vivax*[67,17,23]. Pfs25, the negative control used in this study was a gift from Gavin Wright (Addgene plasmid # 110991; http://n2t.net/addgene:110991; RRID:Addgene_110991). The synthesised plasmids were amplified, purified, and diluted to 1 mg/ml then mixed in a 1:2.5 ratio with Polyethylenimine (PEI 40 kDa, Polysciences.Inc, #24765-1) used as a transfectant (for transfection in 50 ml, 25 μl plasmid (1 mg/ml) in 2.5 ml and 62.5 μl PEI in 2.5 ml freestyle media were mixed). To biotinylate the recombinant proteins, 1.25 μl *E. coli* BirA ligase plasmid was co-transfected with the plasmid of interest. The DNA/PEI mixture was then incubated for 8 min and added to HEK 293E cells. The cells were cultured in Freestyle 293 media (Invitrogen, #12338-018) supplemented with Geneticin (#10131-027) and 1% foetal bovine serum (Sigma, #F2442) and were maintained in a shaker flask at 125 rpm, 37°C, 5% $CO_2$ and 70% relative humidity. The recombinant protein boundaries of the constructs and the amino acid sequences are listed in Supplementary Table 1 and Supplementary Table 3.

### Western blotting
The constructs cloned in the mammalian cell expression plasmids have an intrinsic signal peptide causing the recombinant proteins to get secreted into the culture supernatant. To confirm the expression of the recombinant proteins, 10 μl of culture supernatant was collected from each transfection and mixed with 3.5 μl of NuPAGE 4X LDS sample buffer and 1.5 μl of sample reducing agent (Invitrogen, #NP0009). The samples were heated in 80°C for 10 min and resolved using SDS PAGE (4–12% Polyacrylamide gel, Invitrogen, #NP0323) followed by Western blotting to Nitrocellulose membrane (Amersham, #10600003), with transfer carried out at 100 V for 60 min at 4°C. Membranes were blocked in 3% BSA in TBST (Tris buffer saline with 0.05% Tween20) for 2 h at room temperature and probed using HRP conjugated anti-His antibody (Proteintech, Ptglab, #HRP66005, 1:20,000) overnight at 4°C. After 3 × 10 min washes with TBST buffer, expressed proteins were detected using ECL chemiluminescent substrate (GE Healthcare, USA) in Biorad ChemiDoc MP™ Imaging System. The volume of culture required was optimised for each construct based on the expression level in the blot.

To assess cross reactivity of PVP01_0000100 with other TRAgs, an equal amount (5 μg) of 14 purified PvTRAgs were run on a single gel, and probed with polyclonal antibodies raised against PVP01_0000100. Pfs25 was used as control, as it shares the biolinker hexa-His tag that is common to all the constructs. The protocol for the blotting was largely

as above, except blocking was carried out in 5% milk TBST at room temperature (RT) for 2 h followed by overnight incubation at 4 °C with Rabbit IgGs against the respective TRAgs (1:5000). Blots were washed for 3 × 10 min with TBST, then incubated with HRP conjugated anti rabbit secondary antibody (1:20,000) for 45 min. Following three further washes, antibodies were detected using ECL chemiluminescent substrate (GE Healthcare, USA) in a Biorad ChemiDoc MP™ Imaging System.

## Protein purification

To purify proteins from supernatant, HEK 293E cells were removed by pelleting at 3000 g for 20 min followed by filtering (0.22 μm, Corning). Culture supernatant was adjusted to 25 mM Imidazole and 300 mM NaCl, then loaded on to a 1 ml HiTrap Ni column (Cytiva) pre-equilibrated with binding buffer (40 mM Imidazole, 20 mM Sodium Phosphate, 500 mM NaCl, pH 7.4) at 1 ml/min flow rate. After completion of sample loading, the column was washed with 5 column volume of binding buffer to release any non-specifically bound protein. Elution was performed using step gradients with elution buffer containing 400 mM Imidazole, 20 mM Sodium Phosphate, 500 mM NaCl, pH 7.4. The eluted fractions containing recombinant protein were collected and pooled together for buffer exchange in PBS and concentration in 10 kDa Vivaspin Ultracel concentrators. The final concentrations were measured by Nanodrop OneC UV–Vis Spectrophotometer (Thermo Fisher) and kept in 4 °C until further use.

## Immunisation and IgG purification

Polyclonal IgGs were developed in New Zealand White Rabbits (Eurogentec, Belgium) as per the company's protocol. In brief, 0.8 mg of purified recombinantly *P. vivax* TRAg protein was injected in two rabbits (0.4 mg each) with 100 μg/immunisation/rabbit. Freund's complete adjuvant was used for first immunisation followed by Freund's incomplete adjuvant in booster doses given on days 14, 28 and 58. The final bleed was collected on day 87.

Total IgG was purified from the final bleed rabbit sera using Protein G Gravitrap (GE Healthcare, Cytiva). The sera were diluted as 1:5 in binding buffer (20 mM Sod. Phosphate, pH 7.4). The purification was done using Ab purification kit and the protocol was followed as per the kit's manual. In brief, the diluted sera were loaded in the pre-equilibrated Gravitrap column and the eluant reloaded three times to enrich the binding and the beads washed with 10 ml of binding buffer. The IgG was finally eluted with 5 ml of elution buffer (0.1 M glycine-HCl, pH 2.7) poured into the column and collected directly in a 10 kDa centrifugal filter unit (Millipore) containing 0.225 ml of neutralisation buffer (1 M Tris-HCl, pH 9.0). The IgGs were buffer exchanged three times in RPMI 1640 media (Gibco) for any follow up assay. The final concentration of the purified IgG was determined using a Nanodrop one UV-Vis Spectrophotometer (Thermo Fisher).

## Reticulocyte isolation

**Enrichment of CD71⁺ reticulocytes.** Fresh peripheral whole blood (withdrawn within 48 h) provided by NHS Blood and Transplant (ethical approval University of Cambridge REC HBREC.2019.40 and NHS REC 20/EE/0100). Whole blood was centrifuged for 7 min at 1500 g and the plasma and the white cell layer aspirated, then resuspended with equal volume of phosphate-buffered saline buffer, PBS (Gibco, UK). The solution was passed through a Plasmodipur filter to remove leucocytes (EuroProxima, #8011 Filter 10U) and washed twice with PBS at 1000 g for 10 min. The leucocyte-depleted red blood cell pellet was then resuspended in 10 ml PBS and 5 ml were carefully layered on 6 ml of 70% (v/v) isotonic Percoll cushion (GE Healthcare, #17089101). After centrifugation at 1200 × g for 15 min, brakes off, a thin band(s) of reticulocytes was collected at the Percoll interface and spun down at 300 g for 10 min. All the following steps were carried out at 4 °C to preserve reticulocytes. Microscopic counting of

reticulocytes stained with new methylene blue (NMB, Sigma-Aldrich, #R4132), and the Percoll separation achieved purity of 7–20% reticulocytemia, depending on the blood sample, as previously reported[68]. To purify immature CD71⁺ at purities up to 90%, we used CD71 magnetic MicroBeads (Miltenyi Biotec, # 130-046-201) and followed the company protocol. We confirmed the high purity through sub vital staining (NMB) and microscopy. Reticulocytes were stored at 4 °C and used for binding experiments within 24 h. A binding assay was also performed only with percoll enriched retics without CD71⁺ Ab enrichment to rule out any non specific binding with Anti CD71⁺ coupled beads.

## Reticulocyte binding assay

Purified biotinylated proteins were multimerised on streptavidin coated Nile Red fluorescent beads (Spherotech Inc, 0.52 μm, 0.1% w/v, binding capacity 1.65 nM Biotin-FITC to 1 mg beads). The beads were sonicated to disperse them evenly using a bath sonicator for 10 min with a pulse duration 30 s and 10 s interval between each pulse, all carried out at 4 °C. The sonicated beads were then blocked using 0.2% BSA buffer (in PBS). In a 96 U bottom plate (Nunc 165306, Thermo), 1 μl bead was incubated with 1 μg protein per well for 45 min at 4 °C. Unbound protein was then removed by washing the beads by centrifugation at 3000 g for 10 min at 4 °C. Purified/CD71 enriched reticulocytes were mixed with equal number of erythrocytes (from the same blood sample, using material pelleted under the Percoll cushion) and prestained with Thiazole Orange (10 mg/ml stock solution, 1 μl from the stock added in 10 ml of 0.2%BSA buffer). From this mix 0.5 million cells were added to the washed protein coated beads per well of 100 μl reaction volume and incubated for 45 min at 4 °C. The plate was centrifuged at 250 g for 10 min and the supernatant containing unbound protein was removed carefully. The cell pellet was then resuspended gently with 0.2% BSA buffer and was centrifuged for 10 min at 250 g, this washing step was repeated three times. The cells were finally resuspended in 200 μl 0.2% BSA and analysed using flow cytometry (BD Fortessa). Data were acquired using the FACS Diva software and post processing was performed using FlowJo (ver 10.2). PvRBP2b$_{169-813}$ and Pfs25$_{23-193}$ proteins were used as positive and negative controls respectively. Nile red and Thiazole Orange emissions were detected using 561 and 488 nm lasers respectively. The voltage settings for the respective lasers were FSC-367, SSC-293, 640/670-500, 488-433 and 561/582-480. The gating strategy has been shown in Supplementary Fig. 6. The NR positive binding populations for both retics and erythrocytes has been calculated in terms of percentage with respect to the total retics (TO positive population) and total erythrocytes respectively.

## Structural methods

**Protein cloning, expression and purification.** All protein structural work was conducted prior to the availability of AlphaFold2. To identify constructs best suited for structural analysis, secondary structure composition for PVP01_0000100 was predicted using NetSurf 3.0[69] identifying potential domain boundaries. This identified a C-terminal domain (CTD, D459-693L) of PVP01_0000100 possessing high helical content that was selected for expression and crystallisation studies. The *PVP01_0000100* CTD was PCR amplified from the plasmid containing the *PVP01_0000100* ectodomain (ECD, K62-693L) construct and cloned into the pHLSec vector with AgeI/KpnI restriction sites. The construct was expressed in 200 ml of HEK293F cells (Thermo #R79007) and purified with Ni-NTA (HiTrap, Cytiva) followed by buffer exchange in size exclusion buffer (20 mM Tris,150 mM NaCl, pH 7.4) for three times in a concentrator, MWCO 10 kDa (Milipore) prior to size exclusion (Superdex 75 16/600, analytical Hi load column, Cytiva). The protein was eluted and collected fractions were pooled together and concentrated using a centrifugal concentrator, 10 kDa MWCO (Millipore, Fischer scientific) to 10 mg/ml for crystallisation trials.

**Crystallisation and structure determination.** Initial crystallisation trials were performed in 96-well plates using sitting drops composed of 200 nL protein plus 200 nL reservoir solution. Initial crystals were identified in broad matrix screens (PACT PREMIER, Molecular Dimensions) and diffraction-quality crystals grew in 0.1 M SPG, pH 4.0, 25% (w/v) PEG1500. The crystals were cryo-protected with 20% (v/v) glycerol added as supplement to the reservoir solution prior to flash-freezing in liquid nitrogen. Data were collected at the Diamond Light Source I04 beamline. Data processing and scaling was performed using Xia2 DIALS[70]. Phaser[71] was used for molecular replacement using a model generated from RoseTTAFold[72]. ArpWarp[73] was used to build an initial model of PVP01_0000100 CTD, followed by manual building in Coot[74]. Structure refinement was carried out with Coot[74], ISOLDE[75] and phenix.refine[76]. Data collection and refinement statistics are shown in Supplementary Table 4. Illustrations were generated using The PyMOL Molecular Graphics System, Version 2.0 Schrödinger, LLC. The surface charge analysis for the structure of PVP01_0000100 CTD was carried out with Adaptive Poisson -Boltzmann Solver (APBS) programme (server.poissonboltzmann.org)[77]. The results were analysed in UCSF chimera[78]. The atomic coordinates and structure factors have been deposited in the Protein Data Bank, www.pdb.org under accession code 8ARL.

**Lipid dot blot.** The membrane lipid strips (Echelon P-6002) were blocked with 3% fatty acid free BSA (Sigma, A4612) overnight at 4 °C. The blocked membrane was incubated with 1 µg of PVP01_0000100 CTD, PVP01_0000100 ECD and Pfs25 (negative control) for 2 h at RT. The proteins were incubated overnight followed by three washes with PBST. The strips were then incubated with anti-His-rabbit HRP 1:5000 (Proteintech, Ptglab, USA). Detection was performed with an ECL reagent (GE Healthcare, USA).

**Liposome binding assay.** Phosphotidylcholine (PC) was purchased from Merck (#840051 C). Sulfatide (#131305 P), 1,2-dimyristoyl-sn-glycero-3-phosphoethanolamine-N-(lissamine rhodamine B sulfonyl) (Rhod-PE) (#810157 P) and phosphatidylserine (PS) (#840029 P) were purchased from Avanti®. Lipids were dissolved/suspended in chloroform and the final molar mixtures added to a glass vial before the chloroform was evaporated under a nitrogen stream. To all liposomes prepared, 2% Rhod-PE was incorporated to aid with liposome visualisation and PC was the carrier lipid, with percentages of sulfatide or PS varying from 0 to 80%. The resultant lipid film was hydrated using 50 mM HEPES (pH 7.4) and 150 mM NaCl to form multilamellar liposomes. A final liposome concentration of 0.8 mM in 50 µL was incubated with 1 µM protein for 30 min at room temperature with rotation. The liposomes were then pelleted via centrifugation at 21,000 g and the supernatant removed. The pellet was washed twice before resuspension in 10 µL of 50 mM HEPES (pH 7.4) and 150 mM NaCl and 10 µL of 2X loading dye and then ran on a NuPAGE™ 4–12% Bis-Tris Gel (Invitrogen, #NP0335) and stained with InstantBlue® (Abcam, #ISB1L). Densitometric quantitation of the PS and sulfatide titration assay was performed using ImageJ and GelAnalyzer and then graphed in Prism 9.

**In vitro culture of *P. knowlesi*.** Human O+ erythrocytes were used for the parasite culture and were purchased from NHSBT. The use of human cells for this work was approved by the NHS Cambridge South Research Ethics Committee (REC reference[20]/EE/0100) and by the University of Cambridge Human Biology Research Ethics Committee (HBREC.2019.40). *Plasmodium knowlesi* A1-H.1 were grown in RPMI 1640 medium supplemented with Albumax (Gibco,# 041-96698 A), L-Glutamine (Gibco,# 25030-024), 10% Horse serum (Gibco,# 26050-70), and Gentamicin (Gibco,# 15750-045). The cultures were kept at 37 °C with 2% haematocrit and kept in an incubator containing gas

mixture of 3% $CO_2$, 1% $O_2$ and 96% Nitrogen. The cultures were monitored thrice a week by counting parasitaemia using light microscopy with media change or splitting as appropriate[39].

## Genetic modification of *P. knowlesi* parasites
**PVP01_0000100 knockout construct design and assembly.** The constructs, guide RNAs and primers were designed in Benchling. The 800 bp fragments immediately 5′ and 3′ upstream of *PVP01_0000100* were amplified using *P. knowlesi* genomic DNA obtained from Pk infected erythrocytes using a DNA blood kit (Qiagen) according to the manufacturer's protocol. The eGFP gene was amplified from a plasmid which is used in transfection as positive control. The 5' and 3' homology regions (88 and 85 ng/µl), eGFP gene (40 ng/µl) and pUC19 EcoR1/HindIII digested vector (70 ng/µl) were then added to the Gibson assembly mix at a 1:1:1 ratio and incubated for 1 h at 50 °C.

**Cas9 vector and gRNA assembly.** The cloning vector pKU6-Cas9*ccdB* (1 µg) was digested and dephosphorylated using Fast Digest *Sap*I (*Lgu*I) (Thermo, FD1934) and Alkaline phosphatase (Thermo, EF0654) and ran on a 1% agarose gel followed by purification using agarose gel purification kit (Macherey and Nagel). The Guide RNAs were synthesised from Sigma and reconstituted by mixing 10 µM of the forward and reverse strands for each guide with 1 µl of 10X ligation buffer (NEB), 0.5 µl T4 polynucleotide kinase (NEB) and 6.5 µl nuclease free PCR water. Annealing was carried out by incubating at 37 °C for 30 min, then increasing to 95 °C for 5 min before cooling at 25 °C at a ramp speed of 0.1 °C/sec. Annealed primers were then diluted to 1 µl in 200 µl and ligated to the digested and dephosphorylated pKU6Cas9*ccdB* (50 ng/µl). Two gRNAs were constructed for knocking out *PVP01_0000100* orthologue *PKNH_1300500*. The details of the reaction mixtures and primers for repair template and gRNA vector assemblies are provided in the Supplementary Table 5.

The assembled pUC19 Repair template and gRNA/Cas9 vector were transformed in *E. coli* Ultracompetent cells (XL-10 gold, Agilent) and plated onto Amp+ LB plates. Colonies positive for the insertion were sequence verified (Genewiz) before preparation using a midiprep purification kit (Macherey and Nagel) for transfection.

**P. knowlesi synchronisation, schizont enrichment and transfection.** Synchronization was performed by enriching late-stage parasites using Histodenz (Sigma Aldrich,#D2158).The parasites were centrifuged at 1500 g, and the pellet was resuspended in 5 ml complete media and then layered on top of 5 ml of 55% Histodenz (prepared by adding 2.25 ml of complete medium + 2.75 ml of 100% Histodenz) in a 15 ml tube (Greiner). The mixture was then centrifuged for 10 min at room temperature, 1500 g, acceleration three and brake one, resulting in late-stage parasites becoming enriched at the interface[79]. For immunofluorescence assays and transfections, this was repeated over three consecutive cell cycles to create tightly synchronised parasites, with schizont samples from a fourth cycle of Histodenz purification used for transfection. For each transfection, 20–25 µl of pure late schizonts were mixed with 90 µl of P3 solution in a cuvette (Lonza) containing 20 µg of repair template and 10 µg each of two guide vectors (total 40 µg). Transfections were carried out using programme FP158 (Amaxa Nucleofector, Lonza), and the contents were immediately transferred into a 1.5 ml sterile Eppendorf tube containing 500 µl of prewarmed complete culture media and 150 µl of uninfected erythrocytes (50% Hct). The transfection mix was incubated at 37 °C while shaking at 800 rpm in a thermomixer for 30 min, before being transferred into a six well plate at 37 °C for one parasite life cycle. After 24 h, the parasites were subjected to 100 nM pyrimethamine (Sigma, #P4200000) selection media. The parasites were kept under selection for 7 days with drug media changed every 2 days. On day 7, the drug media was replaced with complete media. The smears of the parasites

were checked for their recovery and parasitaemia around 1%. The parasite cultures were collected for gDNA extraction using a DNA Blood kit (Qiagen) for subsequent genotyping PCR to confirm editing. To clone out the WT population, limiting dilution and plaque cloning in flat-bottomed 96-well plates were performed. Wells containing single plaques were identified using an EVOS microscope (4× objective, transmitted light), expanded, and DNA isolated and genotyped as described above.

**Growth rate assays for genetically modified lines.** Wild type and genetically modified *P. knowlsei* strains were synchronised as described above. Tightly synchronised ring parasites for both the strains were maintained in six-well plates in normal complete RPMI media. Every alternate day, a 100 μL aliquot was collected, centrifuged and thickly smeared on the slide, before being fixed with methanol, air dried and stained with Giemsa for no longer than 10 min. Cells were counted manually under the microscope with 100× oil immersion objective. A total of ~5000 cells were counted for each slide for each line. The parasitaemia were determined and remaining culture was cut down to 1% parasitaemia and grown again. Data were collected this way for 14 days with 7 time points collected. The assay was done twice with technical replicates and analysed by plotting cumulative parasitaemia over time.

**Reticulocyte invasion assays.** A 20 ml culture of *P. knowlesi* were synchronised as described above and schizonts were purified and washed in complete media. A 0.2 μl aliquot of schizonts from PkWT and *PKNH_1300500* KO ( ~90% pure) were added separately to two different tubes containing 4 μl (100% Hct) of enriched reticulocytes purified using CD71$^+$ coated magnetic beads as described above. The final parasitaemia was ~4.5% and the haematocrit was adjusted to 2% by adding 196 μl complete media (total volume 200 μl). A 50 μl volume of culture was added to each well and a set of three wells were set up for the PkWT and *PKNH_1300500* KO line. The plate was incubated at 37 °C for 12 h to ensure at least one round of invasion. The next day, the cells were fixed in 100 μl of 2% PFA, 0.008% GA in PBS for 30 min at 4 °C followed by two times wash with PBS at 450 g, 3 min each. The cells were then permeabilised for 10 min at RT with 0.3% Triton X-100 in PBS followed by three washes with PBS. The cells were treated with 10 μg/ml Ribonuclease A in PBS per 100 μl (per well) for 1 h at 37 °C followed by three washes with PBS. Finally, the cells were stained with SyBr green (1:5000) in PBS at 37 °C for 30 min followed by one wash with PBS at 450 g for 3 min and finally resuspend in 100 μl PBS for analysis using flow cytometry (BD Fortessa).

**P. vivax ex vivo culture and immunofluorescence slide preparation.** A cryopreserved Brazilian *P. vivax* isolate was thawed and the samples were then enriched on 1.080 g/ml KCl high Percoll gradients[80]. Briefly, 2 ml of parasitised red blood cells resuspended using Iscove's modified Dulbecco's medium (IMDM) were layered on 3 ml 1.080 g/mL KCl high Percoll gradient and spun for 15 min at 1200 *g* with slow acceleration and no break. The interface post Percoll-gradient spin was removed, washed in incomplete IMDM, and subjected to in vitro culture conditions with 1% haematocrit in IMDM (Gibco) containing 10% AB$^+$ heat-inactivated sera and 50 μg/ml gentamicin at 37 °C in 5% $CO_2$, 1% $O_2$, and 94% $N_2$[81]. For immunofluorescence assays, slide cytospins were made after 42–44 h of maturation, fixed in ice-cold methanol for 10 min, air-dried and stored at −20 °C.

**Immunofluorescence assays.** To generate *P. knowlesi* slides for immunofluorescence, parasites were synchronised as described above and schizonts purified by density gradient centrifugation using a Histodenz (Sigma) cushion. They were then incubated with the Protein Kinase G inhibitor Compound 2 (1 μM) to prevent egress[82], resulting in the majority of parasites arresting as

extreme late stage mature schizonts. After an hour of this treatment, the media was replaced with normal media for half an hour and finally washed with PBS and fixed for 30 min in 4% paraformaldehyde/0.0075% glutaraldehyde in PBS at room temperature. Fixing buffer was removed by PBS washing, and a schizont smear prepared and fixed on a glass slide. The smear was surrounded by hydrophobic ink from a PAP pen (Daido Sangyo, Japan) to allow low volume staining and washing. For permeabilization, the schizonts were incubated in permeabilization buffer (0.1% Triton X100 in PBS) for 30 min at room temperature. To prevent nonspecific antibody interaction, the cells were blocked overnight at 4 °C using a blocking buffer (3% BSA in PBS or 10% Goat Sera in PBS). For colocalisation studies (both for Pk and Pv), purified rabbit IgGs (described in the IgG purification section) against the selected TRAgs were used (in 1:750 dilution) along with rat PkMSP$_{1–19}$ (in 1:2000 dilution)[18] With these primary antibodies, goat anti rabbit Alexa Fluor 633 (Thermo scientific, #A21070) and goat anti rat Alexa Fluor 488 (Thermo Scientific, #A11006) antibodies were used as secondary antibodies respectively in 1:500 dilutions. Primary antibodies were incubated with the samples sequentially for 1 h each at room temperature. All the secondary antibodies and counterstain Hoechst 33342 (Invitrogen, #H3570) together were incubated with the sample for 45 min at room temperature in the dark. In between each incubation, washing was performed using PBS buffer with 0.05% Tween20 (3 times, 10 min each). All primary and secondary antibodies were prepared in blocking buffer for incubation. After a final wash, the excess washing buffer was removed completely from the smear and the cells were left to mount with a mounting medium (Prolong gold antifade agent, Thermo Fisher) under a coverslip to cure overnight in the dark. The slides were either stored or visualised using a confocal microscope (LSM880 with airyscan setting). The captured images were processed using Zen Blue (2.0) software. The colocalization of PkMSP$_{1–19}$ and the TRAGs antibodies were measured by determining the Pearson correlation coefficient (*r*) using additional JACoP in ImageJ.

**Purification of *Plasmodium knowlesi* merozoites and Immunofluorescence assay.** *P. knowlesi* merozoites were purified as decribed previously[83]. In brief, *Plasmodium knowlesi* A1H1 culture was grown up to a parasitaemia around 9%. The cuture was synchronised three times with 55% Histodenz as described above. After the final synchronisation of the day 3 synchronised schizonts (24 h life cycle stage of the parasite on the final day), the top layer of the schizonts was collected and washed with 10 ml RPMI complete media. The schizont pellet was resuspended in RPMI complete media containing Compound 2 (1 μM) for 2 h to block schizont rupture and allow merozoite maturation[84]. The cells were then washed to remove Compound 2 and finally resuspended in the same for 30 min at 37 °C in 3% $CO_2$, 1% $O_2$, and 96% $N_2$ to revive the parasites. The cells were first passed thorugh 3 μM (Nucleopore etched membrane, Whatman 110612) and then immediately through 2 μM filters (Nucleopore etched membrane, Whatman 110611). The eluant, containing free merozoites (schizonts can not fit through the filters) was directly collected into the fixing buffer. The fixed cells were centrifuged at 8801 *g* for 5 min, resuspended in PBS and stored at 4 °C. The free *P. knowlesi* merozoites (2 μl) were smeared on to glass slides. One set was permeablilsed with permeabilisation buffer (0.1% Triton X100 in PBS) for 30 min at room temperature and the other set was remained non-permeabilised. The cells were washed with PBS to remove traces of Triton X-100; this was used as washing buffer all throughout the experiment. The cells were blocked overnight at 4 °C using a blocking buffer (3% BSA in PBS). The colocalization studies were performed as described for the *P. vivax* schizont staining with same

dilutions of the primary and secondary antibodies. The slides were visualised using a confocal microscope (LSM880 with Airyscan setting). The captured images were processed using Zen Blue (2.0) software.

**Flickering spectrometry of reticulocytes coated with proteins.** 50 µl of CD71 + reticulocytes were used as control and incubated at 4 °C for 45 min in PBS + 0.2% BSA with 100 µg/ml Pfs25 (negative control), and 100 µg/ml PVP01_0000100, respectively. Samples were washed once and diluted in PBS supplemented with 0.2% BSA at 0.01% Hct and transferred in a Secure Seal Hybridisation Chamber (Sigma-Aldrich, UK) attached to a glass cover slip; the same was kept at 37 °C temperature through a heated collar. Cell membrane thermal fluctuations were recorded at a high frame rate (514 frames per s) and short exposure time (0.8 ms) for 20 s with a 60X Plan Apo VC 1.40 NA oil objective using a Nikon Eclipse Ti-E inverted microscope (Nikon, Japan). Videos were acquired in bright-field with a red filter using a CMOS camera (model GS3-U3-23S6M-C, Point Grey Research/FLIR Integrated Imaging Solutions (Machine Vision, Ri Inc., Canada) (Supplementary Movies 1–3). The equatorial contour for each frame was tracked by an in-house Matlab programme as previously described[44,85].

The radial component of each contour is then decomposed into its Fourier modes, and the average power spectrum for each cell is:

$$\left\langle \left| \tilde{u}(q_x) \right|^2 \right\rangle = \frac{k_B T}{L} \frac{1}{2\sigma} \left( \frac{1}{q_x} - \frac{1}{\sqrt{\frac{\sigma}{\kappa} + q_x^2}} \right) \tag{1}$$

$u$ is the amplitude of fluctuations, $q_x$ is the mode, brackets denote averaging across all contours for a cell, $k_B$ is the Boltzmann constant, $T$ is the temperature, and $L$ is the mean circumference of the cell contour. $\sigma$ and $\kappa$ are the tension and bending modulus, respectively, for which the same programme fits the average power spectrum for mode numbers between 5 and 20, as done in ref. [86]. Average mean-square amplitude spectra for each condition are shown in Supplementary Fig. 14. Higher modes are dominated by noise, whereas lower modes are dominated by the shape of the cell. Recent work has reported the effects of out of plane fluctuations for vesicles[87] that has not been implemented yet on red blood cells.

**Cell membrane wrapping energy during invasion.** The wrapping energy was calculated by considering changes to the free energy of the erythrocyte membrane due to a localised deformation and depends on its tension and bending modulus. We consider the deformation to be localised to a disc of radius $R$ which is deformed by a merozoite to a half-sphere of the same radius. Assuming a very simple hemi-spherical geometry, the sum of these two contributions is given by

$$\Delta F = \pi R^2 \sigma + 4\pi\kappa \tag{2}$$

where $R = 1\,\mu m$ is the radius of the merozoite[88,89], $\sigma$ is the tension, and $\kappa$ is the bending modulus. The above equation is derived in ref. [44].

**Reporting summary**
Further information on research design is available in the Nature Portfolio Reporting Summary linked to this article.

## Data availability
The authors declare that the data supporting the findings of this study are available within Main Text, Supplementary Figures, Tables and videos. Additional data generated during the peer review process are provided in the peer review file. The structure factors and atomic coordinates for PVP01_0000100 C terminal domain are deposited in the PDB under the accession code 8ARL. Source data are provided with this paper.

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

## Acknowledgements

We would like to thank Prof. Gavin J. Wright for his help during the construct designing for expression vectors and for providing the HEK 293E cell line for protein expression. Dr. Robert W. Moon for providing the CRISPR Cas9 plasmid and *Plasmodium knowlesi* A1.H1 line. Dr. Reiner Schulte, Gabriela Grondys-Kotarba and Chiara Cossetti of the CIMR Flow Cytometry facility for providing required training and assistance during the flow cytometry experiments. We also like to convey our thanks to Matthew Gratian and Mark Bowen for providing training and data acquisition in LSM880 confocal microscope with Airyscan setting. We are grateful to Alison Kemp for her guidance to use the CRISPR Cas9 system. Ellen Knuepfer for the kind donation of Rat anti-PkMSP1-19 serum. We thank Stephen Graham for help with crystallographic model building. We acknowledge Diamond Light Source for time on beamline I04 under proposal MX21426. This work was funded by the National Institutes of Health (R01AI137154) and the Wellcome Trust (220266/Z/ 20/Z). J.E.D. and S.J.M. are supported by a Wellcome Trust Senior Research Fellowship (219447/Z/19/Z) awarded to J.E.D. As this research was funded in part by the Wellcome Trust, for the purpose of open access, the author has applied a CC BY public copyright license to any Author Accepted Manuscript version arising from this submission.

## Author contributions

P.K., J.C.R., J.E.D. and M.D. conceived and designed the research; P.K., D.N., S.M., V.I., performed the experiments, P.K., D.N., S.M., V.I., P.C., M.D., J.E.D. and J.C.R. analysed the data; P.K and J.E.D. performed structural refinement and analysis, M.U.F., M.D., S.D. and U.K. provided the Pv isolates for the IFA studies. P.K. J.C.R., D.N., J.E.D., S.M. wrote the first draft of the manuscript and all authors provided input.

## Competing interests

The authors declare no competing interests.
