## [Peer Review File · Nature Communications]

The structure of a Plasmodium vivax Tryptophan Rich Antigen domain suggests a lipid binding function for a pan-Plasmodium multi-gene familyREVIEWER COMMENTS

Reviewer #1 (Remarks to the Author):

In this well-written manuscript, Kundu et al present an important new set of data about the TRAGs family of Plasmodium surface proteins. They show the TRAGs to be localised at the merozoite surface. They show that reticulocyte binding capability is found in the C-terminal domain. They solve the structure of one such domain, showing analogy to a BAR domain and show that this domain binds to specific lipids. Finally, they show that TRAGs knockout can reduce reticulocyte invasion. This is an interesting and well-conducted set of studies which provides substantial new insight into this protein family.

I found the early parts of this study, including the IFAs, reticulocyte binding analysis and structural studies, very convincing. However, I was less convinced by the lipid binding experiments. My understanding is that BAR domains do not have specificity for individual lipids, with specificity instead mediated by neighbouring domains which bind to specific head groups. Instead, BAR domains mediate electrostatic interactions which drive or recognise curvature. Based on the structure, wouldn't one expect the same for the TRAGs? The authors do some experiments in Figure 4 to show specificity for particular lipids, but I wasn't entirely convinced. In the 'Sphingostrip' experiment in Figure 4B, how is the lipid attached to the strip? Is it in large liposomes, which would be representative of membrane surfaces, or disassembled into single lipids? If the later, then this is not how the lipid would be seen in the reticulocyte by the TRAGs. The liposome experiment in Figure 4C seems more convincing, but there were a few issues. Firstly 50% sphingolipid seemed rather a lot to represent the amount on a red cell surface. Secondly, could the difference between this lipid mix and the PC liposomes be due to charge alone? Perhaps the authors could do a similar experiment using high concentration of PS, which, like the sphingolipid is negatively charged. If they see a big difference between the sphingolipid and the PS liposomes, then they could more confidently invoke specificity rather than just charge.

The authors next knock out one of the TRAGs and assess the effect on invasion. In Figure 5D, there is no significant effect. However, in Figure 5E, they do see a significant effect, just for reticulocytes. [Incidentally, it seemed to me that the organisation of Figure 5E was not right.

Shouldn't the indicators for significance be for WT vs KO for each of reticulocytes and erythrocytes? i.e. a graph order of WT then KO for retics and then WT then KO for erythrocytes.] They see a change in tension and also a change in wrapping energy. In the case of wrapping energy, the Pfs25 control is lacking. It would be good to add this in, in case transfection or selection had an effect. Otherwise, this data is interesting, and it is actually quite surprising that, for a large protein family, a significant effect is seen for KO of just one of these proteins.

In summary, this is an interesting and well conducted study. My main concern relates to the lipid specificity experiments, which in my view need some more work. Otherwise, the paper should be published with minor changes, and will be the most comprehensive available analysis of this important protein family.

Minor comments:

1. The labelling on the tree in Figure 1A was not legible and the labels need to be larger.
2. The authors use IFA to assess whether the TRAGs are surface localised. They raise antibodies and assess their specificity, with most antibodies shown through testing against a TRAGs panel to show specific binding. These antibodies were then used for co-localisation experiments with a merozoite surface protein and colocalization is observed. They authors claim that this shows the protein to be 'on the merozoite surface' (line 21 or 116). My understanding is that this study was conducted using permeabilised cells and therefore, strictly, the authors cannot be sure whether they see proteins on the surface, or they are on the inner surface. If equivalent experiments have not been done with non-permeabilised merozoites, then the experiment does not prove surface exposure and the authors should re-write to clarify the finding.
3. AlphaFold predictions of a couple of other examples suggested that the fold is well conserved. While this is uncontroversial, would it be possible to do this more systematically – choosing examples for AlphaFold from different locations on the evolutionary tree of TRAGs to assess the strength of this conclusion across a wide sequence range?
4. I think that the authors need to think about how they describe the relationship to the BAR domain proteins. They say that TRAGs is a 'BAR domain fold' (line 223) or that it has 'homology' (line 21) or 'similarity' (line 259). These three things are different and I

wondered if it is really a BAR domain, or is not related to a BAR domain but is analogous in being a three helical curved bundle. A curved three-helical bundle architecture is not so rare, and so the authors should decide on their terminology – is it a BAR domain, or something which looks similar to a BAR domain through convergent evolution? My guess is the later. Something built from three curved helices and analogous to a BAR domain in function, but not a BAR domain.

Reviewer #2 (Remarks to the Author):

The authors generate antibodies against 4 TRAg family members of *P. vivax*. These show localization to the merozoite surface, and one of these they can validate with a knockout. They furthermore recombinantly express and purify several (>15) of these TRAg proteins and test their ability to bind to reticulocytes and erythrocytes. They study one of these more carefully, and show that the tryptophan-rich domain of this PVP01_000010 can bind sulfatides. By knocking out the gene for this protein, the authors show that this protein contributes to enhanced invasion specifically of reticulocytes by *P. knowlesi*, although the authors do not know if sulfatides are enriched in reticulocytes relative to erythrocytes. The data shown for this protein is of great interest. They suggest that the tryp-thr-rich domain of these proteins is analogous to lipid-binding BAR domains, which is a very intriguing hypothesis, and they have some data to support it, but there are some loose ends. In particular, what can be concluded about all members of the TRAg family vs what can be concluded about one or more particular members of the TRAg family gets muddled throughout the manuscript.

Line 146-148. This sentence overreaches. The authors have not shown secretion of all TRAg, so calling all that are annotated as not secreted an “artifact of automated genome annotation” seems unjustified. What they can say from this experiment is that at least 2 proteins with no predicted SP are engaged by the secretory pathway, and they can speculate that others may be as well.

Line 172: How are the “ectodomains” determined? Both in terms of what portion of the protein is expressed, And how do the authors know the orientation of the proteins?

Can the PVP01_0000100 antibody recognize unpermeabilized *P. knowlesi* merozoites? This would tell us if it is in the right orientation to bind reticulocytes as suggested.

The reticulocytes are isolated with antibodies and beads, which I assume remain associated with the reticulocytes during protein binding. Could the proteins that show interaction with reticulocytes actually be interacting with these beads or antibodies. Can this be easily controlled for? To be fair, at least for PVP01_0000100, binding to these beads would not explain the binding to erythrocytes.

Line 244: The authors show that the three-helical fold is well conserved across orthologs in three species, but is it also conserved across paralogs? Do the Tryptophan/Threonine-rich domains from other TRAGs that did not bind reticulocytes also have this structure? Or is this structure unique to this protein with the unique RBC-binding properties. Either outcome is interesting. The way the paper is written currently often suggests the former, but their RBC binding data in figure 2 suggests all these proteins have different binding abilities, so I am curious what is supported by structural analysis.

Line 303: “suggesting that each member of the TRAg family may have specificity towards different lipids” (also lines 478-484) (i) the *P. falciparum* protein domain seems to have a mostly negatively charged surface and I thought it was the positive charges that promoted the lipid interactions of BAR domains. Is it safe to imply that this domain across the TRAg family acts like a BAR domain, when one of the three analysed does not have this characteristic of a BAR domain? (ii) I thought these three proteins in figure S8 were considered orthologs, but this sentence (line 303, but also the discussion) seems to be referring to other TRAg members within one species. This brings us back to the question above of how these structural results relate to other TRAGs within a species. Maybe knowing this information might help support the statement made here. The idea that different TRAGs bind different lipids is an attractive hypothesis, and it is one that the authors can test. Although I mentioned the structural analysis above, even better: the authors have expressed recombinant TRAg proteins - couldn't one or more of those be used for the lipid binding assays? Do these TRAGs bind lipids? Different lipids?

Lines 398-415: This is a lot of assumptions based on localizing two of the proteins in the family for which the SP is not annotated. Also, proteins containing a shared domain can have differing localizations. Line 400 suggests that all the proteins in the family have the same function, but even the RBC-binding data in this manuscript here do not support that conclusion.

Throughout the discussion, conclusions based on experiments with this one TRAg are expanded to include multiple TRAgS, e.g., line 461; line 465. Their RBC binding data show different TRAg proteins have different binding capacities, but then they go on to suggest that their data about the one TRAg will apply to (all?) others.

Line 472: "suggesting that many lipids, including sulfatide, are likely to be more abundant in reticulocytes" Of course certain lipids could get enriched in this process, but some, presumably also sulfatide, could also be reduced in this process. You can only say the lipid content of the reticulocyte plasma membrane is likely to be different from that of erythrocytes.

Line 217: it took me a while to figure which region is considered the C-terminal domain. And the term "full length" suggests the whole protein with the predicted hydrophobic region was expressed and purified (which seemed surprising). One item in the "structural methods" described full length as "(FL, K62-693L)" suggested that "full-length" is not really the whole protein, but this is really not clear in the text. This search made me realize that the protein regions and the sequences used for all of the proteins expressed and used in Figure 2 are absent (at least I didn't find them) and should be included.

Minor points:

Fig 1A is not readable.

Supplementary Figure. 1: I miss the size indications or ladder on the blots. And the resolution is not ideal.

Supplementary Figure 2: PVP01_0948700 appears to me more intracellular than MSP1 or than the other TRAgS. Is this the case in most parasites stained with this antibody?

Line 185: It is not clear from the figure which 5 proteins show significant binding. Only 2 are obvious.

Figure 2B: is the y axis % of cells?

Why is C-terminal portion better at binding RBCs than the “full length” protein?

If the trypt-thr regions are similar, and the rest of the protein inhibits binding, could the TRAg domain from other proteins show binding to reticulocytes if expressed by themselves?

Line 283: “whereas Pfs25 didn’t show any significant binding to either” I am not sure this statement is entirely accurate, and I suggest rephrasing.

Line 267: “suggesting suggests”

Reviewer #3 (Remarks to the Author):

I would be commenting on the flickering spectroscopy part of this work. In summary, for in vivo cells in this study, it is hard to reconcile if such analysis can be done at the length scales of 2-4 um cells where the microscopy reaches theoretical resolution limits to determine flickering accurately as high as 20th modes. The authors have not provided videos of contour detection or flickering spectra to accurately prove that indeed they detect those spectra or not some white noise. Specific comments regarding this big issue is dispersed in the points below. These points would help to make the method concrete and conclusions reliable.

1) Can the authors provide movies of fluctuations with contour detection of the cell for both control, negative and positive cases? It is astounding to see that the authors can perform such analysis on 2-4 um cells with such accuracy. I have huge doubts that the authors can detect such fluctuations with great confidence interval. A rough calculation of the focal

depth (z resolution) using λ/NA^2 where λ is (wavelength) assumed 514 nm and for 1.40 NA is ~ 250 nm. For lateral XY resolution, if using 514 nm, with NA of 1.40, condenser with an NA of 0.95, then the (theoretical) limit of resolution will be 261 nm based on Rayleigh Criteria. The fluctuations reported would barely meet the criteria. The authors should provide the details in the main text as this is critical. A recent work from Dimova group, *Soft matter* 16, 8996-9001, 2020, demonstrated, for large $\lambda = \text{Focal Depth}/\text{Radius} > 0.15$, vesicles demonstrate softening effects due to out of plane fluctuations with both simulations and experiments (Supplement S6). The authors should cite this work and report the value they obtain here.

2) For the spectrum, what scaling is observed, do they follow, q^{-1} or q^{-3} ? Is it a bending dominated spectra or tension dominated?

3) Why is the Helfrich spectrum determined with experiments with such high acquisition speeds? The expression used for average variance considers independent experiments meaning independent fluctuation shape modes. With such high fps, the modes would be correlated, and the statistics might be affected. Please comment. Has the fluctuations reached equilibrium?

4) The authors can also determine the autocorrelation functions of fluctuation amplitudes to see the differences due to the added agents with the same data they have. The experiments are at high fps so that should be doable. Is the scaling consistent (Tension or Bending rigidity dominated) with Helfrich.

Response to Reviewer's Comments

We thank the Reviewers for taking their valuable time to comment on our paper. We have provided a point-by-point response to each of these comments below, together with details of how we have performed further experiments that address these points and how they have been incorporated into a revised version of the manuscript.

Reviewer #1

In this well-written manuscript, Kundu et al present an important new set of data about the TRAGs family of Plasmodium surface proteins. They show the TRAGs to be localised at the merozoite surface. They show that reticulocyte binding capability is found in the C-terminal domain. They solve the structure of one such domain, showing analogy to a BAR domain and show that this domain binds to specific lipids. Finally, they show that TRAGs knockout can reduce reticulocyte invasion. This is an interesting and well-conducted set of studies which provides substantial new insight into this protein family.

We would like to convey our sincere thanks to Reviewer #1 for taking time to evaluate our manuscript. Kindly find our point-by-point response to the questions given below.

I found the early parts of this study, including the IFAs, reticulocyte binding analysis and structural studies, very convincing. However, I was less convinced by the lipid binding experiments. My understanding is that BAR domains do not have specificity for individual lipids, with specificity instead mediated by neighbouring domains which bind to specific head groups. Instead, BAR domains mediate electrostatic interactions which drive or recognise curvature. Based on the structure, wouldn't one expect the same for the TRAGs?

Although some of the most well-characterised F-BAR-containing proteins do appear to primarily interact with membranes via relatively non-specific electrostatic interactions this is not the case across all of the diverse BAR-domain containing family of proteins. Several BAR domains have been shown to possess specific differences in their lipid binding properties². For example yeast Rvs161/167 specifically binds membranes containing PI(4,5)P2 while other BAR-containing proteins do not bind this lipid³. However, as detailed below we have now explored this key question of specificity further using additional liposome binding assays.

The authors do some experiments in Figure 4 to show specificity for particular lipids, but I wasn't entirely convinced. In the 'Sphingostrip' experiment in Figure 4B, how is the lipid attached to the strip? Is it in large liposomes, which would be representative of membrane surfaces, or disassembled into single lipids? If the later, then this is not how the lipid would be seen in the reticulocyte by the TRAGs.

Sphingostrips are membranes that have been spotted with 15 different lipids at 100 pmol per spot. The reviewer is correct that these commercially available strips are not prepared using liposomes, instead the lipids are dissolved in an appropriate organic solvent and spotted on to the nitrocellulose, meaning the lipid head groups are randomly arrayed rather than presented in a specific orientation as they would be in a membrane. However, these strips do provide a useful first-pass method for rapidly testing binding to a large number of lipids for subsequent follow-up using specific lipids incorporated into more physiologically-relevant liposomes. This is the approach we have carried out in this study.

The liposome experiment in Figure 4C seems more convincing, but there were a few issues. Firstly 50% sphingolipid seemed rather a lot to represent the amount on a red cell surface.

The use of 50% sulfatide in the binding assays is based on the knowledge that within membranes, sphingolipids are locally concentrated within membrane microdomains on the

cell surface. This enrichment results in very high local concentrations of the sphingolipid headgroups. However, as detailed below we have now repeated these experiments at a range of sulfatide (and PS) concentrations to demonstrate specific binding.

Secondly, could the difference between this lipid mix and the PC liposomes be due to charge alone? Perhaps the authors could do a similar experiment using high concentration of PS, which, like the sphingolipid is negatively charged. If they see a big difference between the sphingolipid and the PS liposomes, then they could more confidently invoke specificity rather than just charge.

We thank the reviewer for this excellent suggestion and have now carried out a series of experiments using PS as suggested to explore whether binding is primarily driven by charge (Fig. R1). We began initially by using liposomes with either 50% sulfatide or 50% PS and tested binding of Pfs25, the negative control, PVP01_0000100_CTD, and NF155, the positive control. We found that Pfs25 binding didn't change notably with the addition of sulfatide or PS to the liposomes. NF155 displayed binding to sulfatide but not PS, mirroring our recently published data with this protein, while PVP01_0000100_CTD showed preferred binding to both liposomes containing sulfatide and PS. To further explore this, we titrated the PS or sulfatide percentages in the liposomes from 0-80%. Densitometric analysis of this data demonstrated that PVP01_0000100_CTD binding to sulfatide fit well to a sigmoidal binding curve (indicative of a specific binding interaction). However, binding to PS was not so clear. PS shows an initial linear increase in binding (suggestive of non-specific binding), followed by a decrease in binding at the higher percentages. Although it is complicated to interpret exactly what this means, these data suggest that while there is some binding to PS, it appears to be a non-specific charge effect that does not possess a typical binding curve as seen for sulfatide. Enhancement of liposome binding of BAR domain proteins in the presence of PS has been seen previously² supporting our argument that TRAGs possess BAR-like properties. The nature of the binding curve to sulfatide provides even more compelling evidence that we are

Figure R1. Protein binding to liposomes of different compositions. (A) Liposomes were composed as follows. PC: 98% phosphatidylcholine (PC), 2% Rhodamine-phosphatidylethanolamine (Rhod-PE), S: 48% PC, 2% Rhod-PE, 50% sulfatide, PS: 48% PC, 2% Rhod-PE, 50% phosphatidylserine (PS). Top panel is an example of the raw data (SDS PAGE) and below is the densitometric analysis across 3 replicates. (B) Titration of sulfatide and PS in liposomes from 0-80%. All liposomes have 2% Rhod-PE and PC makes up the remainder. Top panel is an example of the raw data and bottom panel is the densitometric analysis of the SDS PAGE gels across 3 replicates (+/- SEM).

demonstrating a specific interaction between PVP01_0000100_CTD and the sulfated-galactose headgroup of sulfatide. Due to the importance of this difference in binding of PVP01_0000100_CTD to these two negatively charged lipids, we have now included these data in the manuscript, and discuss our interpretation of this result and the implications for the specificity of sulfatide binding.

The authors next knock out one of the TRAGs and assess the effect on invasion. In Figure 5D, there is no significant effect. However, in Figure 5E, they do see a significant effect, just for reticulocytes. [Incidentally, it seemed to me that the organisation of Figure 5E was not right. Shouldn't the indicators for significance be for WT vs KO for each of reticulocytes and erythrocytes? i.e. a graph order of WT then KO for retics and then WT then KO for erythrocytes.] They see a change in tension and also a change in wrapping energy. In the case of wrapping energy, the Pfs25 control is lacking. It would be good to add this in, in case transfection or selection had an effect. Otherwise, this data is interesting, and it is actually quite surprising that, for a large protein family, a significant effect is seen for KO of just one of these proteins.

We would like to thank Reviewer 1 for the suggestion. Figure 5E is now updated with the data reorientated as suggested. We also repeated the biophysical experiments using Pfs25 as a negative control as requested, and this data is included In Figure 5F.

In summary, this is an interesting and well conducted study. My main concern relates to the lipid specificity experiments, which in my view need some more work. Otherwise, the paper should be published with minor changes, and will be the most comprehensive available analysis of this important protein family.

We again would like to thank Reviewer 1 for their constructive comments which have helped immensely to improve this manuscript.

Minor comments:

1. The labelling on the tree in Figure 1A was not legible and the labels need to be larger.

We agree that the size of Figure 1A was suboptimal. Figure 1 has now been changed to include additional immunofluorescence experiments requested in point 2 below. To accommodate this and address the reviewer's advice, we have also moved what was Figure 1A into a new Supplementary Figure 1, where it can take up a full page size, allowing us to use a larger font size for the gene IDs to increase the clarity of the phylogenetic tree.

2. The authors use IFA to assess whether the TRAGs are surface localised. They raise antibodies and assess their specificity, with most antibodies shown through testing against a TRAGs panel to show specific binding. These antibodies were then used for co-localisation experiments with a merozoite surface protein and colocalization is observed. They authors claim that this shows the protein to be 'on the merozoite surface' (line 21 or 116). My understanding is that this study was conducted using permeabilised cells and therefore, strictly, the authors cannot be sure whether they see proteins on the surface, or they are on the inner surface. If equivalent experiments have not been done with non-permeabilised merozoites, then the experiment does not prove surface exposure and the authors should re-write to clarify the finding.

We completely agree that this is an important point and thank Reviewer 1 for pointing it out. We have now performed IFA with and without permeabilization, using purified *P. knowlesi* merozoites (such experiments would be impossible with *P. vivax ex vivo* isolates, where the volume of parasite material is always extremely limited). All four anti-TRAG antibodies showed strong staining both with and without permeabilization and co-localise with MSP1. By contrast, merozoites stained with an antibody against the Inner Membrane Complex (IMC) marker

protein MyoA Interacting Protein (MTIP) was only able to stain its target in the presence of permeabilization, not in its absence. The IMC sits closely underneath the merozoite surface and can be confused with it using IFA in late schizonts with permeabilization. This new data is now included in the manuscript as Figure 1B (MTIP staining with/without permeabilization) and Figure 1C (TRAg staining without permeabilization) and we believe unequivocally confirms the surface localisation of all four TRAgS, regardless of the signal peptide prediction. We are grateful to the reviewer for suggesting experiments for solidifying this important point.

3. AlphaFold predictions of a couple of other examples suggested that the fold is well conserved. While this is uncontroversial, would it be possible to do this more systematically – choosing examples for AlphaFold from different locations on the evolutionary tree of TRAgS to assess the strength of this conclusion across a wide sequence range?

Again, this was a very helpful suggestion, thank you. In response, we have explored the extent of conservation of the TRAg fold more systematically both within the *P. vivax* gene family, and across the *Plasmodium* genus. Within *P. vivax*, we have selected four PvTRAgS (PVP01_1033800, PVP01_1401800, PVP01_0202200 and PVP01_1101400) distributed across the phylogenetic tree as suggested, which range in amino acid sequence identity of the C-terminal domain from 28 to 42% relative to PVP01_0000100, and generated AlphaFold models of the C-terminal domain for all of them. Superposing the C α atoms of these models onto our experimentally determined PVP01_0000100 structure indicate a very high level of structural conservation, with RMSD values ranges from 1.2-2 Å as shown in **Figure R2** below.

Figure R2. Structural homology of distantly related TRAg domains within *P. vivax*. Four different TRAgS were selected based on their position in the phylogenetic tree and with sequence identity ranging from 28-42%. The C-terminal domains for each of these four TRAg domains were modelled using AlphaFold2 and superposed onto the experimentally determined structure of PVP01_0000100 using Coot to generate backbone RMSD values listed in the table.

In addition, we compared the TRAg domain of PVP01_0000100 with the TRAg domain from 7 different species of *Plasmodium* including all common model species and all human-infective species. In each case, the specific TRAg was selected based on it having the lowest sequence identity compared to the PVP01_0000100 CTD. All domains were modelled using AlphaFold 2, and RMSD values calculated, shown in **Figure R3** below. The only domain that exceeded an RMSD of 3 was the TRAg domain from *P. malariae*. This strongly supports that the overall

fold of TRAg domains is highly conserved across the entire *Plasmodium* genus, with the main regions of difference being short helices and loops at the distal ends of the extended helical core. This is despite the % amino acid identity being relatively low, focussed on conserved tryptophans which are buried in the interior of the fold.

These more extensive comparisons are now included in the manuscript, with these figures included in the supplement.

Figure R3. Structural conservation of the BAR like domain in TRAg across different *Plasmodium* species. Seven different TRAgS were selected from different *Plasmodium* species for structural comparison. TRAgS were selected based on having the lowest sequence identity to the TRAg domain of PVP01_0000100 CTD and were modelled with AlphaFold 2. The structures were superposed with the experimentally determined structure of PVP01_0000100 CTD and the RMSD values were based on the number of residues aligned as shown in the table.

4. I think that the authors need to think about how they describe the relationship to the BAR domain proteins. They say that TRAgS is a 'BAR domain fold' (line 223) or that it has 'homology' (line 21) or 'similarity' (line 259). These three things are different and I wondered if it is really a BAR domain, or is not related to a BAR domain but is analogous in being a three helical curved bundle. A curved three-helical bundle architecture is not so rare, and so the authors should decide on their terminology – is it a BAR domain, or something which looks similar to a BAR domain through convergent evolution? My guess is the later. Something built from three curved helices and analogous to a BAR domain in function, but not a BAR domain.

We thank the reviewer for this point and we agree these are important distinctions to make. In response, we have clarified the text such that throughout we refer to TRAg domains as possessing structural homology to BAR domains rather than designating them as BAR domains. We therefore now refer to them as possessing a helical fold with structural homology to BAR domains. We agree that a three-helical curved bundle is not so rare, but note that the structural homology was considered significant using search engines such as DALI (<http://ekhidna2.biocenter.helsinki.fi/dali/>) and the functional homology was confirmed in our liposome-binding experiments which have now been expanded in the new manuscript. Whether these proteins have structural and functional homology to BAR domains via convergent evolution remains unclear and will be interesting to explore in future work.

Reviewer #2

The authors generate antibodies against 4 TRAg family members of *P. vivax*. These show localization to the merozoite surface, and one of these they can validate with a knockout. They furthermore recombinantly express and purify several (>15) of these TRAg proteins and test their ability to bind to reticulocytes and erythrocytes. They study one of these more carefully, and show that the tryptophan-rich domain of this PVP01_000010 can bind sulfatides. By knocking out the gene for this protein, the authors show that this protein contributes to enhanced invasion specifically of reticulocytes by *P. knowlesi*, although the authors do not know if sulfatides are enriched in reticulocytes relative to erythrocytes. The data shown for this protein is of great interest. They suggest that the try-p-thr-rich domain of these proteins is analogous to lipid-binding BAR domains, which is a very intriguing hypothesis, and they have some data to support it, but there are some loose ends. In particular, what can be concluded about all members of the TRAg family vs what can be concluded about one or more particular members of the TRAg family gets muddled throughout the manuscript.

We would like to convey our sincere thanks to Reviewer #2 for taking time to evaluate our manuscript. Kindly find our point-by-point response to the questions given below.

Line 146-148. This sentence overreaches. The authors have not shown secretion of all TRAGs, so calling all that are annotated as not secreted an “artifact of automated genome annotation” seems unjustified. What they can say from this experiment is that at least 2 proteins with no predicted SP are engaged by the secretory pathway, and they can speculate that others may be as well.

We agree that this sentence was inelegantly worded, and that we have only provided positive evidence that two PvTRAGs are secreted despite lacking a predicted signal peptide (SP). We believe though that the central point is still worth making. Genome annotation is frequently more automated than many readers may realise, and this is particularly true for the *P. vivax* genome, which is unquestionably not as well annotated as that of *Plasmodium falciparum*. We have found that all of the PvTRAGs contain a hydrophobic region at the N terminus, which for some TRAGs are predicted as signal peptide by the SignalP server (the only server used in PlasmoDB database for SP prediction) but for others where SignalP does not predict an SP, a SP is positively predicted by other SP prediction servers (Phobius and PrediSi). Even when no server predicts an SP, the hydrophobic region may still act as an SP, as the unusual AT-bias and amino acid composition of *Plasmodium* genomes may not always serve as good templates for SP servers based largely on predictions from much more GC-rich model eukaryotic organisms. We have adjusted the working within the relevant section to try to avoid any sense of over-reaching, while still making the central point, which we believe is an important limitation of automated genome annotation and one that is likely relevant to other genes.

Line 172: How are the “ectodomains” determined? Both in terms of what portion of the protein is expressed, And how do the authors know the orientation of the proteins? Can the PVP01_0000100 antibody recognize unpermeabilized *P. knowlesi* merozoites? This would tell us if it is in the right orientation to bind reticulocytes as suggested.

Ectodomains were defined as the entire sequence from the end of a predicted SP (or the hydrophobic N-terminal region mentioned above) until a predicted membrane-anchoring domain, such as TMD or GPI consensus motif. As TRAGs all lack predicted membrane anchoring domains, in reality that means we expressed the entirety of the protein downstream of the predicted SP/hydrophobic region. The transmembrane domains were predicted using TMHMM 2.0 (<https://services.healthtech.dtu.dk/services/TMHMM-2.0/>) and Phobius

(<https://phobius.sbc.su.se/>). The signal peptides were predicted using a SignalP (<https://services.healthtech.dtu.dk/services/SignalP-3.0/>), Phobius and PrediSi (<http://www.predisi.de/>).

To answer the second part of the question, we thank Reviewer 2 for the suggestion. As noted in response to a question from Reviewer 1, we have now tested all four antibodies against unpermeabilised purified *P. knowlesi* merozoites and found that the merozoite surface was clearly stained and colocalised with merozoite surface staining marker MSP1 in all cases, whereas the IMC marker (MTIP) control only stained in the presence of detergent, allowing access to the intracellular IMC organelle. Staining for unpermeabilised *P. knowlesi* merozoites with anti-PVP01_0000100 are shown below in **Figure R4**, which we believe confirm that the ectodomain is exposed on the surface of the merozoite in the right orientation to bind reticulocytes. This data is now included in the main figure as Figure 1B and 1C.

Figure R4. Immunofluorescence in the absence of detergent. Non-permeabilised purified *P. knowlesi* merozoites were co-stained with anti-PVP01_0000100 and anti-PkMSP1 antibodies to confirm merozoite surface localisation.

The reticulocytes are isolated with antibodies and beads, which I assume remain associated with the reticulocytes during protein binding. Could the proteins that show interaction with reticulocytes actually be interacting with these beads or antibodies. Can this be easily controlled for? To be fair, at least for PVP01_0000100, binding to these beads would not explain the binding to erythrocytes.

This is an important question, and we are grateful to the reviewer for raising it. To address it, we repeated the binding experiments using reticulocytes purified using only a 70% Percoll cushion. This results in lower yield and purity (which is why we didn't use it originally) but avoids the use of anti-CD71 beads. Thiazole Orange (TO) staining (which detects RNA, which is absent from mature erythrocytes) estimated the proportion of reticulocytes post-Percoll enrichment as 4.50%, lower than the yield after CD71⁺ Ab beads enrichment as performed in the manuscript, but still enough to explore binding. PVP01_0000100 coated beads resulted in a shift of 33% of the total reticulocyte population, whereas the negative control, Pfs25, shifted only 5% of the total reticulocyte population. The positive control PvRBP2b shifted 66% of the reticulocyte population, showing stronger binding as expected (see **Figure R5**). We believe that this experiment confirms that the proteins are not binding to the anti-CD71 beads in any way but are clearly binding to the reticulocytes. We have included this new data as a Supplementary Figure.

Figure R5: Red blood cell binding assay performed with Percoll enriched reticulocytes. The reticulocytes rRNA's get stained with Thiazole orange (TO) whereas the erythrocytes remain unstained. The respective population (reticulocyte or erythrocyte) bound to protein coated streptavidin Nile red (NR) beads will shift to Q2 or Q3 quadrant (Q1: TO+/NR-, Q2: TO+/NR+, Q3: TO-/NR+ and Q4: TO-/NR-).

Line 244: The authors show that the three-helical fold is well conserved across orthologs in three species, but is it also conserved across paralogs? Do the Tryptophan/Threonine-rich domains from other TRAGs that did not bind reticulocytes also have this structure? Or is this structure unique to this protein with the unique RBC-binding properties. Either outcome is interesting. The way the paper is written currently often suggests the former, but their RBC binding data in figure 2 suggests all these proteins have different binding abilities, so I am curious what is supported by structural analysis.

We thank the Reviewer for this comment and agree that a better description of the extent of structural conservation across TRAg domains is needed. The three helical fold is strongly conserved across *P. vivax* paralogs and across *Plasmodium* species, as described in the response to Reviewer 1 and shown in Figures R2 and R3 above. Indeed, the three-helical fold is confidently predicted by AlphaFold2 for all TRAg C-terminal domains (CTDS) we have explored to date, despite some CTDS possessing primary sequence identity as low as 19%.

Importantly, this structural homology is defined by the fold of the protein backbone which is partially driven by the highly conserved tryptophan residues that lie buried within the three-helical bundle. This means that the primary sequence variability contributes to changes in the surface properties of the different TRAg domains, but not to the fold itself. These surface differences can be illustrated by comparing the surface hydrophobic and electrostatic properties of a number of PvTRAGs, which show considerable changes despite the same backbone fold (see **Fig R6** below). It is these surface properties that are likely to contribute to the differential binding to RBCs noted by the reviewer and may also be involved in defining lipid-binding specificity.

The next logical question therefore is if these other TRAGs are not binding RBCs, what are they doing? Given the strong conservation of the fold as a whole, and further evidence provided below, we believe that it is likely that lipid binding is conserved across many TRAGs, and it is possible that some of the lipids that individual TRAGs interact with are not presented on the outer leaflet of red blood cells. The question of whether lipid-binding by TRAGs are involved in other steps in the parasite life cycle, or in other biological processes such as RBC remodelling post-invasion is a fascinating one, but one that we believe is beyond the scope of this manuscript. We now make that possibility clearer in the manuscript and hope that the central finding, that members of this neglected protein family have lipid-binding properties via a fold that is conserved across all members (which we have now confirmed more extensively, as shown below) is an important step forward for the field to explore these kinds of questions.

Figure R6. Comparison of PvTRAg surface properties. A) Three *P. vivax* TRAGs were selected based on experimentally determined RBC binding data, and the AlphaFold2 models of their C-terminal domains (CTD) superimposed with the experimentally determined structure of the CTD of PVP01_0000100, emphasising the conservation of the core fold. B) Table comparing the primary sequence identity, the extent of fold conservation as indicated by the RMSD values (1.3-1.48 Å) and the alignment of the number of C α atoms. PVP01_0503700, the other PvTRAg with clear RBC binding activity in our data, has the highest level of primary sequence identity to PVP01_0000100, but the difference is only between 42.0% and 49.0% identity, suggesting identity alone is not likely to drive RBC binding. C) By contrast, the surface charge (red: negatively charged and blue: positively charged) and hydrophobicity (yellow: most hydrophobic through white, to dark cyan: most hydrophilic) differed more significantly between TRAGs and could relate to the difference in their ability to bind the red blood cell surface.

Line 303: “suggesting that each member of the TRAg family may have specificity towards different lipids” (also lines 478-484) (i) the *P. falciparum* protein domain seems to have a mostly negatively charged surface and I thought it was the positive charges that promoted the lipid interactions of BAR domains. Is it safe to imply that this domain across the TRAg family acts like a BAR domain, when one of the three analysed does not have this characteristic of a BAR domain?

We thank the reviewer for this point. Based on this and a Reviewer #1 comment we are being more cautious in our use of the term BAR domain throughout the text, now referring to the structural homology with BAR domains and describing BAR domain-like membrane binding properties of TRAGs rather than designating these domains as *bona fide* BAR domains, which the Reviewer rightly points out we have not confirmed. However, as detailed above, although the overall fold is conserved the surface properties can be substantially different supporting that the lipid-binding properties may be different or absent, see our response and figure below.

(ii) I thought these three proteins in figure S8 were considered orthologs, but this sentence (line 303, but also the discussion) seems to be referring to other TRAg members within one species. This brings us back to the question above of how these structural results relate to other TRAGs within a species. Maybe knowing this information might help support the statement made here. The idea that different TRAGs bind different lipids is an attractive hypothesis, and it is one that the authors can test. Although I mentioned the structural analysis

above, even better: the authors have expressed recombinant TRAg proteins - couldn't one or more of those be used for the lipid binding assays? Do these TRAGs bind lipids? Different lipids?

We agree with the reviewer that it would strengthen the central case to test lipid binding for multiple members of the PvTRAg family. In response, we have selected extracellular domains of four PvTRAGs for further binding assays, chosen based on their expression and purification profile (**Figure R7A**). The levels of primary amino acid identity to PVP01_0000100 range from 19-31% (PVP01_0202200- 31%, PVP01_0404200- 29%, PVP01_0948700-19%, PVP01_1401800-30%). Liposome binding assays were conducted to test binding to PC and sulfatide as well as incorporating the suggestion from Reviewer #1 to include PS as a test of non-specific electrostatic interactions. All five PvTRAGs demonstrated some binding to liposomes but possessed different intensities depending on the liposome composition (**Figure R7B**) – two PvTRAGs preferentially bound sulfatide-containing liposomes (PVP01_0000100 and PVP01_0948700) while the other three do not show substantial binding preference for sulfatide. Based on these data and the conservation of structural homology to BAR domains, it seems reasonable to propose that other TRAGs will also bind lipids,

Figure R7: Different binding specificities of TRAg domains for liposomes with different lipid compositions. A: The five PvTRAGs were affinity purified followed by size exclusion chromatography. Some contaminants can be seen via SDS PAGE, but proteins were deemed pure enough for liposome binding assays. B: Binding of the 5 full length PvTrags to liposomes containing either PC alone, 50% sulfatide (S) or 50% PS. NF155 which specifically binds sulfatide was used as positive control ¹.

but with different specificities. Exploring the full repertoire of lipids for these different TRAg proteins is beyond the scope of this paper but is actively being pursued as ongoing work in our labs. We have adapted the relevant section in the manuscript and incorporated this new data as Supplementary Figure.

Lines 398-415: This is a lot of assumptions based on localizing two of the proteins in the family for which the SP is not annotated. Also, proteins containing a shared domain can have differing localizations. Line 400 suggests that all the proteins in the family have the same function, but even the RBC-binding data in this manuscript here do not support that conclusion.

We agree with the reviewer that we were perhaps overly enthusiastic in how we presented the implications of our study. We believe that this manuscript represents two important steps forward – that some PvTRAGs without predicted SPs can be clearly secreted, and that those same PvTRAGs can bind lipids via a protein fold that we have experimentally confirmed, and that AlphaFold2 strongly predicts is present in all family members. We therefore hypothesise that all TRAGs are secreted and may bind lipids, but of course we have only shown this for four PvTRAGs, so this is absolutely a hypothesis only. We have rewritten the relevant section of the manuscript to make this more clear, and to state that lipid binding is an important function to test for other TRAGs, rather than being proven.

We certainly agree that other TRAGs may have different functions and may be localised in different places. Certainly this seems to be true in *P. berghei*, where recent data suggests that PbTRAGs are exported into the infected erythrocyte, where they may be involved in RBC remodelling⁴. We would note that this function is entirely consistent with lipid binding activity (as RBC remodelling involves the creation of membrane-bound compartments within the RBC), even if this has not been shown for PbTRAGs at this stage, but of course it does not absolutely require it. However, it is worth pointing out that the TRAg family has undergone dramatic expansion in *P. vivax* and related parasite species, and it may be risky to extrapolate function in these species from either *P. berghei* or *P. falciparum*, which have only 3-5 TRAGs and may represent an ancestral function.

We have adjusted the Discussion to try to make these points and raise these hypotheses, which we believe are important and useful for the field, but without trying to over-state what can be extrapolated from our data.

Throughout the discussion, conclusions based on experiments with this one TRAg are expanded to include multiple TRAGs, e.g., line 461; line 465. Their RBC binding data show different TRAg proteins have different binding capacities, but then they go on to suggest that their data about the one TRAg will apply to (all?) others.

As noted above, we agree with the Reviewer that we were too definitive in our interpretation about function in the original Manuscript. We have adjusted this section to make it clear that it is quite possible/even likely that some PvTRAGs are involved in other biological processes than RBC binding and invasion, and that what is true in *P. vivax* (and perhaps most likely closely related species where the TRAg family has expanded dramatically) may not necessarily be true in other more distantly related species such as *P. falciparum* and *P. berghei*.

Line 472: “suggesting that many lipids, including sulfatide, are likely to be more abundant in reticulocytes” Of course certain lipids could get enriched in this process, but some, presumably also sulfatide, could also be reduced in this process. You can only say the lipid content of the reticulocyte plasma membrane is likely to be different from that of erythrocytes.

We agree with the Reviewer and have adjusted this section.

Line 217: it took me a while to figure which region is considered the C-terminal domain. And the term “full length” suggests the whole protein with the predicted hydrophobic region was expressed and purified (which seemed surprising). One item in the “structural methods” described full length as “(FL, K62-693L)” suggested that “full-length” is not really the whole protein, but this is really not clear in the text. This search made me realize that the protein regions and the sequences used for all of the proteins expressed and used in Figure 2 are absent (at least I didn't find them) and should be included.

The information regarding length of the constructs that have been trialled for expression in this manuscript are mentioned in supplementary Table 1. We also have now also included the information in the main text of the manuscript. The amino acid sequences of all the constructs used in the study are now included in the supplementary file as requested.

Minor points:

Fig 1A is not readable.

We agree. As noted in the response to Reviewer 1, we have moved the phylogenetic tree to a new Supplementary Figure, where we could increase its size, and increase the font size of the Gene IDs. We hope that this combines with the high quality image that allows for zooming in, makes it more readable.

Supplementary Figure. 1: I miss the size indications or ladder on the blots. And the resolution is not ideal.

Thank you for pointing this out, which was an oversight. The resolution has now been increased, and sizes/ladder included as suggested.

Supplementary Figure 2: PVP01_0948700 appears to me more intracellular than MSP1 or than the other TRAGs. Is this the case in most parasites stained with this antibody?

The staining with anti PVP01_0948700 antibody is indeed not very clear, perhaps partly because they are rabbit polyclonal antibodies, so it is possible that they have some off-target binding. As noted above, we have now repeated staining for all antibodies using unpermeabilised purified *P. knowlesi* merozoites, which gives an improved staining quality for PVP01_0948700 and confirms that the protein is extracellular (**Figure R9**: bottom panel, the images were taken in 100X oil Immersion objective in Leica LSM 880 with AiryScan, scale bar 2 μ M).

Figure R9. Staining of permeabilised *P. knowlesi* schizonts (top panel) compared purified non-permeabilised purified *P. knowlesi* merozoites (bottom panel) with anti PVP01_0948700 antibodies. MSP1 used as surface staining marker. Scale bar 2 μ M

Line 185: It is not clear from the figure which 5 proteins show significant binding. Only 2 are obvious.

We agree with the reviewer that this was not clear. To take a more systematic approach, we defined “significant binding” as binding above the level seen in the negative control (Pvs25) + 2x standard deviations. PVP01_0000100 and PVP01_0503700 are both very clearly above this cut-off and we refer to these as ‘strong binders’, while are PVP01_0949200, PVP01_0700900 and PVP01_1469900 all also show binding above this cut-off but are much weaker. Figure 2 and its legend have been amended to make this clear

Figure 2B: is the y axis % of cells?

The Y axis of Figure 2B is now more clearly labelled as “% RBC bound to protein coated strep beads”.

Why is C-terminal portion better at binding RBCs than the “full length” protein? If the tryp-thr regions are similar, and the rest of the protein inhibits binding, could the TRAg domain from other proteins show binding to reticulocytes if expressed by themselves?

We think this is possibly because the N-terminal regions (which are predicted to be unstructured) may be partially impeding the well folded C terminal domain and preventing it from accessing lipids. We would emphasise though that the full-length does still bind both RBCs and lipids. As to whether other TRAg domains could bind reticulocytes if expressed alone, this is an interesting question, and are keen to do in future studies. However, we have now included data that shows that multiple ECD’s of PvTRAGs can bind lipids, again suggesting that if the N-terminal region of the protein is impeding binding, it is only doing so partially.

Line 283: “whereas Pfs25 didn’t show any significant binding to either” I am not sure this statement is entirely accurate, and I suggest rephrasing.

We have altered this line to read: “Pfs25, the negative control, did not appreciably bind either liposome composition.”

Line 267: “suggesting suggests”

The error is rectified in the main text

Reviewer #3

I would be commenting on the flickering spectroscopy part of this work. In summary, for in vivo cells in this study, it is hard to reconcile if such analysis can be done at the length scales of 2-4 μm cells where the microscopy reaches theoretical resolution limits to determine flickering accurately as high as 20th modes. The authors have not provided videos of contour detection or flickering spectra to accurately prove that indeed they detect those spectra or not some white noise. Specific comments regarding this big issue is dispersed in the points below. These points would help to make the method concrete and conclusions reliable.

We are grateful to Reviewer #3 for taking time to evaluate our manuscript. Kindly find our point-by-point response to the questions given below.

1) Can the authors provide movies of fluctuations with contour detection of the cell for both control, negative and positive cases? It is astounding to see that the authors can perform such analysis on 2-4 μm cells with such accuracy. I have huge doubts that the authors can detect such fluctuations with great confidence interval. A rough calculation of the focal depth (z resolution) using λ/NA^2 where λ is (wavelength) assumed 514 nm and for 1.40 NA is ~ 250 nm. For lateral XY resolution, if using 514 nm, with NA of 1.40, condenser with an NA of 0.95, then the (theoretical) limit of resolution will be 261 nm based on Rayleigh Criteria. The fluctuations reported would barely meet the criteria. The authors should provide the details in the main text as this is critical. A recent work from Dimova group, *Soft matter* 16, 8996-9001, 2020, demonstrated, for large $\lambda = \text{Focal Depth} / \text{Radius} > 0.15$, vesicles demonstrate softening effects due to out of plane fluctuations with both simulations and experiments (Supplement S6). The authors should cite this work and report the value they obtain here.

Movies of fluctuations and contour detection are now provided in the Supplementary Information for all conditions studied (reticulocytes only, reticulocytes + 100 μg of Pfs25, and reticulocytes + 100 μg of PVP01_0000100). Since the original microscope videos are of the order of Gb, we provide instead compressed movies played at 50 frames per second, each image was taken every 10 frames. We believe that from these movies it is still possible to appreciate the cell membrane fluctuations.

The reviewer correctly estimates our optical resolution, which is determined by the experimental setup and measurement conditions and is indeed of the order of 250-300nm. However, the amplitudes we refer to are the amplitudes of flickering modes, which are estimated from the fluctuations observed across the whole cell contour with radius 4 μm . For the highest mode considered $n = 20$ (**Figure R10** and **Supplementary Fig. 14**, $q = 4\mu\text{m}^{-1}$). In general, the wave vector $q = 2\pi n/L$, where L is the perimeter of the cell contour and n the mode number. The fluctuation wavelength is $2\pi/q = L/n = 2\pi/4 \mu\text{m}^{-1} = 1.57 \mu\text{m}$. The fluctuation wavelength is the quantity that need to be higher than the optical resolution (in our case is about 5 times higher) to be able to distinguish fluctuations. For high $q \gg 4\mu\text{m}^{-1}$ ($n \gg 20$) it is not possible to detect fluctuations within our temporal and spatial resolution.

The cell boundary at each frame is here defined within a few nm following the procedure in⁵. Briefly, we first obtain a correlation kernel by calculating the average radial profile of pixel intensity near the boundary at 1° angular increments. With this kernel we calculate the correlation value for every point along the radial direction, find the peak of correlation (which

identifies the boundary point), and repeat this calculation for each angle to construct the full cell contour. Although not as accurate, simpler contour detection methods based on linear gradient fitting of pixel intensity also routinely determine contours within <50nm, still well below the optical resolution of a typical setup⁶.

Intuitively, the higher accuracy in determining mode amplitude can be understood considering that the information to calculate mode amplitudes is collected from a large number of pixels, of the order of Length of contour/ pixel size = $30 \mu\text{m}/0.0986 \mu\text{m} \approx 300$.

We thank the referee for rightly pointing out the work of Dimova's group, and we now include it in our references in the main text. Such work is the continuation of a previous work by Pietro Cicuta's group⁷ which found that out-of-focus fluctuations may contribute to systematic overestimation of the bending modulus. It is difficult to estimate the focal depth using an inverted brightfield microscope as the one used here, and not a confocal one, therefore we can only highlight the fact that we performed all sets of experiments by adjusting the focus to keep the same equatorial plane for all observations, limiting effects due to out of plane fluctuations across different conditions. In our group, we are actively working on implementing such corrections on red blood cells, which is a more complicated system that requires a dedicated work beyond the scope of this manuscript.

2) For the spectrum, what scaling is observed, do they follow, q^{-1} or q^{-3} ? Is it a bending dominated spectra or tension dominated?

The spectra for reticulocytes, (black line) reticulocytes + 100 $\mu\text{g}/\text{ml}$ Pfs25 (blue line), and reticulocytes + 100 $\mu\text{g}/\text{ml}$ PVP01_0000100 (orange line) are now included in **Supplementary Fig. 14** and reported below. Each spectrum is the average \pm SEM for the number of cells analysed (respectively, 65, 73, and 79). Tension clearly dominates the modes from 5 to about 10, and it is in this regime that the difference between fluctuations of reticulocytes + PVP01_0000100 is more evident with respect to the control. The highest modes tend to follow a q^{-2} trend, where possibly both tension and bending modulus play a role.

Figure R10: Fluctuation spectra across conditions for modes 5-20 (reticulocytes in black, reticulocytes coated with 100 $\mu\text{g/ml}$ Pfs25 in blue, and reticulocytes coated with 100 $\mu\text{g/ml}$ PVP01_0000100 in orange). Average spectra (line) and standard deviation (shadowed band) of 65, 73, and 79 cells are shown, respectively. For modes 5- 10 tension dominates, indicated by q^{-1} dashed line, modes 11-20 tend to follow q^{-2} trend.

3) Why is the Helfrich spectrum determined with experiments with such high acquisition speeds? The expression used for average variance considers independent experiments meaning independent fluctuation shape modes. With such high fps, the modes would be correlated, and the statistics might be affected. Please comment. Have the fluctuations reached equilibrium?

We agree that we acquired at high speed (it results by reducing the field of view from 1920x1200 pixel to 256x256) and this can lead to correlated frames. However, fluctuations have reached the equilibrium, and we checked it by plotting mean-square amplitude spectra for the first half and second half of the movie separately (**Figure R11**) for a representative cell for each condition tested. The spectra are indeed overlapping, meaning that the number of frames and the duration of the movie is enough for the fluctuations to reach equilibrium.

Figure R11: The duration of the recorded movies is enough for cell membrane fluctuations to reach equilibrium. Mean-square amplitude spectra for the first half (left) and second half (right) of 3 different representative movies, one per tested condition. Fit for modes 5-20.

4) The authors can also determine the autocorrelation functions of fluctuation amplitudes to see the differences due to the added agents with the same data they have. The experiments are at high fps so that should be doable. Is the scaling consistent (Tension or Bending rigidity dominated) with Helfrich.

We thank the referee for suggesting a complemented approach that considered the dynamics of modes. We have followed the work as described previously^{5,8}. In **Figure R12** we report a) the normalised autocorrelation function, fitted with an exponential function for the mode 6 for a typical single cell data per condition (Videos in the Supplementary); b) relaxation time versus q modes fitted with $\tau_q = 0.8[2(\eta_{int} + \eta_{out}) / (\sigma q + \kappa q^3)]$ for modes 5-20 for the same cells. The scaling is consistent with the tension-dominated regime, as for Helfrich's calculation.

Figure R12: Autocorrelation function of fluctuation amplitudes and relaxation time. a) Normalised autocorrelation function fitted with an exponential function for the mode 6 for a representative cell per condition. b) Relaxation time fitted with static parameters for bending modulus and tension derived by Helfrich equation. The trend line follows the tension regime (q^{-1}).

References

- 1 McKie, S. J. *et al.* Altered plasma membrane abundance of the sulfatide-binding protein NF155 links glycosphingolipid imbalances to demyelination. *Proc Natl Acad Sci U S A* **120**, e2218823120, doi:10.1073/pnas.2218823120 (2023).
- 2 Salzer, U., Kostan, J. & Djinic-Carugo, K. Deciphering the BAR code of membrane modulators. *Cell Mol Life Sci* **74**, 2413-2438, doi:10.1007/s00018-017-2478-0 (2017).
- 3 Zhao, H. *et al.* Membrane-sculpting BAR domains generate stable lipid microdomains. *Cell Rep* **4**, 1213-1223, doi:10.1016/j.celrep.2013.08.024 (2013).
- 4 Gabelich, J. A. *et al.* A member of the tryptophan-rich protein family is required for efficient sequestration of *Plasmodium berghei* schizonts. *PLoS Pathog* **18**, e1010846, doi:10.1371/journal.ppat.1010846 (2022).
- 5 Yoon, Y. Z. *et al.* Flickering analysis of erythrocyte mechanical properties: dependence on oxygenation level, cell shape, and hydration level. *Biophys J* **97**, 1606-1615, doi:10.1016/j.bpj.2009.06.028 (2009).
- 6 Pecreaux, J., Dobreiner, H. G., Prost, J., Joanny, J. F. & Bassereau, P. Refined contour analysis of giant unilamellar vesicles. *Eur Phys J E Soft Matter* **13**, 277-290, doi:10.1140/epje/i2004-10001-9 (2004).
- 7 Rautu, S. A. *et al.* The role of optical projection in the analysis of membrane fluctuations. *Soft Matter* **13**, 3480-3483, doi:10.1039/c7sm00108h (2017).
- 8 Kariuki, S. N. *et al.* Red blood cell tension protects against severe malaria in the Dantu blood group. *Nature* **585**, 579-583, doi:10.1038/s41586-020-2726-6 (2020).

REVIEWERS' COMMENTS

Reviewer #1 (Remarks to the Author):

The authors have responded very constructively to my comments and have provided additional data and experiments, all of which are well-conducted. My only minor comment is that I would have preferred to see the limitations of the sphigostrips described in the manuscript and Figure R1 included as a supplementary figure so that the readers can also see these responses. However, I am happy for the manuscript to be published as it is and congratulate the authors on an important piece of work.

Reviewer #2 (Remarks to the Author):

The additional experiments and re-wording of some key sections improved this already interesting study. In general, it is now much clearer which conclusions relate to one TRAg member vs which relate to all, except in the abstract (which is important):

Line 29: The way the abstract is phrased it sounds as though the tryptophan-rich domain that defines the TRAg family has lipid binding activity with preference for sulfatide. This line should be rephrased to clarify that the domain of PVP01_0000100 has this property. The idea that all of these TRAg domains may bind lipids is a reasonable hypothesis that is supported by the data, but it should be phrased as such if it is kept in the abstract.

(Figure R7/Sup Fig 11 serves its role in this manuscript, because the experiment shows differential binding to sulfatide and *suggests* all tested domains can bind lipids, but care should be taken to not expand to larger conclusions from this data without more controls).

Reviewer #3 (Remarks to the Author):

The authors have successfully addressed my questions with detailed responses. I can now recommend this work to be published.

REVIEWERS' COMMENTS

Reviewer #1 (Remarks to the Author):

The authors have responded very constructively to my comments and have provided additional data and experiments, all of which are well-conducted. My only minor comment is that I would have preferred to see the limitations of the sphingostrips described in the manuscript and Figure R1 included as a supplementary figure so that the readers can also see these responses. However, I am happy for the manuscript to be published as it is and congratulate the authors on an important piece of work.

We extend our sincere gratitude to Reviewer #1 for their valuable time and effort in evaluating our manuscript. Their insightful feedback has significantly enhanced the quality of our work. As recommended in these comments, we have now included a statement in the Results section to highlight the limitations of the sphingostrips, as follows:

Commercial sphingostrips are made by dissolving lipids in organic solvent and spotting them onto a nitrocellulose membrane, resulting in lipid head groups being randomly arrayed rather than presented in a specific orientation as they would be in a membrane. These strips therefore provide a useful rapid screening tool to test binding to a wide range of lipids, but do not necessarily recapitulate physiological lipid binding. To test binding in a more relevant context, the interaction with sulfatide was confirmed using giant multilamellar liposomes composed of 48% phosphatidylcholine (PC), 2% Rhodamine-labelled phosphatidylethanolamine (Rhod-PE) and 50% sulfatide.

Regarding inclusion of Fig R1 as a supplementary figure, in fact Figure R1B was already included within the main figures in the revised manuscript. We believe as per the journal policy the response file will be published alongside the main manuscript, allowing readers to access all of Figure R1 and gain further insights from the peer review process. Once again, we express our gratitude to Reviewer #1 for their invaluable comments and suggestions.

Reviewer #2 (Remarks to the Author):

The additional experiments and re-wording of some key sections improved this already interesting study. In general, it is now much clearer which conclusions relate to one TRAg member vs which relate to all, except in the abstract (which is important):

Line 29: The way the abstract is phrased it sounds as though the tryptophan-rich domain that defines the TRAg family has lipid binding activity with preference for sulfatide. This line should be rephrased to clarify that the domain of PVP01_0000100 has this property. The idea that all of these TRAg domains may bind lipids is a reasonable hypothesis that is supported by the data, but it should be phrased as such if it is kept in the abstract. (Figure R7/Sup Fig 11 serves its role in this manuscript, because the experiment shows differential binding to sulfatide and *suggests* all tested domains can bind lipids, but care should be taken to not expand to larger conclusions from this data without more controls).

We are very much thankful to Reviewer #2 for taking time to evaluate our manuscript that has helped us to improve it. We have now adjusted the abstract as suggested to emphasise that sulfatide binding is associated with PVP01_0000100 specifically, as follows:

Biochemical assays confirm that PVP01_0000100 C-terminal domain has lipid binding activity with preference for sulfatide, a glycosphingolipid present in the outer leaflet of plasma membranes.

Once again, we express our gratitude to Reviewer #2 for their invaluable comments and suggestions.

Reviewer #3 (Remarks to the Author):

The authors have successfully addressed my questions with detailed responses. I can now recommend this work to be published.

We are very much thankful to Reviewer #3 for taking time to evaluate our manuscript.